# Generalization Aware Minimization

## Abstract

Sharpness-Aware Minimization (SAM) optimizers have improved neural network generalization relative to stochastic gradient descent (SGD). The goal of SAM is to steer model parameters away from sharp regions of the training loss landscape, which are believed to generalize poorly. However, the underlying mechanisms of SAM – including whether its bias toward flatter regions is why it improves generalization – are not fully understood. In this work, we introduce Generalization-Aware Minimization (GAM), derived by directly applying the goal of guiding model parameters toward regions of the landscape that generalize better. We do so by showing mathematically through a Bayesian derivation that the landscape of expected true (test) loss is a rescaled version of the observed training loss landscape, and that a sequence of perturbative updates in place of SAM's single perturbative update can optimize the expected test loss. We present a practical online algorithm to implement GAM's perturbative steps during training. Finally, we empirically demonstrate that GAM has superior performance over SAM, improving generalization performance on a range of benchmarks. We believe that GAM provides valuable insights into how sharpness-based algorithms improve generalization, is a superior optimizer for generalization, and may inspire the development of still-better optimizers.

## 1 Introduction

Generalization is a fundamental challenge in training deep neural networks, where the goal is to perform well on unseen data rather than just fitting the training set. One promising approach to enhance generalization is *Sharpness-Aware Minimization* (SAM) (Foret et al., 2021), which has empirically demonstrated success by guiding model parameters away from sharp minima in the training loss landscape. The underlying intuition is that flatter minima correspond to solutions that are less sensitive to perturbations and thus generalize better to new data.

Despite its empirical effectiveness, the theoretical understanding of why SAM improves generalization remains limited. Recent studies have questioned whether SAM's bias toward flatter regions is the primary reason for its success (Wen et al., 2023). This ambiguity highlights the need for a deeper exploration of the mechanisms through which SAM and similar algorithms enhance generalization. Understanding the mechanisms behind SAM is crucial for developing more effective optimization algorithms that consistently improve generalization across various architectures and datasets.

In this work, we adopt a Bayesian perspective to investigate what the observed training loss landscape reveals about the expected test loss. Analytically, we derive a relationship between the training and test loss landscapes under the assumption of general quadratic loss functions, a reasonable assumption in many conditions. Our analysis reveals that the expected test loss landscape is a rescaled version of the training loss landscape. This rescaling result implies that it may be possible to directly optimize the expected test loss, instead of using the indirect hypothesis that promoting flatness is better for generalization.

Building on this insight, we introduce *Generalization-Aware Minimization* (GAM), a generalization of SAM that employs multiple perturbation steps designed to transform the observed training loss landscape to the rescaled expected test loss landscape. GAM moves beyond SAM's heuristic of flatness by directly targeting the expected test loss, thereby enabling better generalization. Moreover, we develop a practical online algorithm that adapts the perturbation sizes during training by using

the training loss on auxiliary minibatches as a proxy for the test loss and demonstrate superior performance.

Our contributions are as follows:

- **Theoretical Insight:** We demonstrate that for quadratic loss functions, the expected test loss landscape is a rescaled version of the observed training loss landscape. This provides a direct link between training dynamics and generalization performance.

- **Gradient Transformation:** We show that the gradient of the expected test loss can be obtained by evaluating the gradient of the training loss after applying a specific sequence of parameter perturbations. This finding bridges the gap between optimizing for training loss and directly targeting test loss.

- **Algorithm Design:** Based on our theoretical results, we propose GAM, an algorithm that extends SAM by using multiple perturbation steps with higher-order derivatives and by tuning perturbation sizes online during training. This makes GAM practical for use in large-scale neural network training. We recover SAM as the one-step perturbation specialization of GAM.

- **Empirical Validation:** We empirically validate GAM on benchmark datasets including MNIST, CIFAR-10, SVHN and ImageNet. Our results show that GAM consistently leads to better generalization than baselines.

## 2 RELATED WORK

### 2.1 SHARPNESS-AWARE MINIMIZATION (SAM) ALGORITHMS

Sharpness-Aware Minimization (SAM) algorithms were introduced to improve the generalization of neural networks by favoring solutions in flatter regions of the training loss landscape, which have empirically been linked to better generalization performance (Foret et al., 2021). SAM perturbs model parameters in the direction of the loss gradient and then optimizes using a second gradient step, effectively minimizing the sharpness of the loss function. Numerous extensions and variants of SAM have since been proposed, focusing on improving computational efficiency and generalization (Mi et al., 2022; Liu et al., 2022a;b; Du et al., 2022a;b; Li et al., 2024; Wu et al., 2024).

Despite its empirical success, the theoretical understanding of SAM remains limited and an active area of research (Andriushchenko et al., 2023; Zhuang et al., 2022; Chen et al., 2024; Si & Yun, 2024; Dai et al., 2024). Recent studies have raised questions about whether SAM's generalization improvements stem directly from its bias toward flatter regions of the loss landscape. For instance, some works argue that SAM's effectiveness may not always be directly attributable to sharpness, but instead to other implicit regularization effects introduced by the perturbation procedure (Wen et al., 2023; Andriushchenko & Flammarion, 2022). Our work builds on this debate by introducing a generalized framework that moves beyond the sharpness heuristic and directly targets the expected test loss.

### 2.2 BAYESIAN OPTIMIZATION

Bayesian optimization is a well-established framework for optimizing functions that are expensive to evaluate, and it has been successfully applied in hyperparameter tuning and low-dimensional optimization problems (Snoek et al., 2012; Frazier, 2018). The fundamental principle of Bayesian optimization is to maintain a probabilistic model of the objective function and update it using new observations, guiding the search toward areas of the input space that are likely to yield better outcomes.

While Bayesian optimization has shown promise in various applications, its applicability to high-dimensional settings, such as neural network training, has been limited. Methods that rely on Gaussian processes or other surrogate models struggle to scale due to the curse of dimensionality and high computational costs. Although some efforts have extended Bayesian optimization to use gradient-based information for more scalable updates (Wu & Frazier, 2016; Wu et al., 2017; Shekhar & Javidi, 2021), these approaches have yet to achieve widespread practical adoption in deep learning beyond hyperparameter optimization.

# 3 GENERALIZATION AWARE MINIMIZATION

In this section, we present our theoretical framework and introduce *Generalization-Aware Minimization* (GAM), a novel optimization algorithm designed to directly improve generalization by aligning the training loss landscape with the expected test loss landscape.

## 3.1 PROBLEM SETUP AND NOTATION

Consider a parametric model with parameters $\theta \in \mathbb{R}^d$. Let $L(\theta)$ denote the true (test) loss function, which measures the expected loss over the data distribution $\mathcal{D}$. In practice, we have access only to the empirical training loss $\tilde{L}(\theta)$ computed over a finite training dataset sampled from $\mathcal{D}$. Our objective is to find the parameter vector $\theta$ that minimizes $L(\theta)$, even though we can only observe and optimize $\tilde{L}(\theta)$.

To formalize our analysis, we consider a quadratic loss function. Note that any general smooth loss landscape can be approximated locally as a quadratic, so we expect our analysis to hold for general losses within a small enough local neighborhood. In what will follow, we will derive an optimization rule assuming that the quadratic approximations are local, and therefore may vary over the course of training. Specifically, we consider the true loss function $L(\theta)$ and the observed training loss function $\tilde{L}(\theta)$ given by:

$$L(\theta) = \frac{1}{2}(\theta - \theta^*)^T M(\theta - \theta^*) + c, \tag{1}$$

$$\tilde{L}(\theta) = \frac{1}{2}(\theta - \tilde{\theta}^*)^T \tilde{M}(\theta - \tilde{\theta}^*) + \tilde{c}, \tag{2}$$

where $\theta^*, \tilde{\theta}^* \in \mathbb{R}^d$ represent the minima of the loss functions, $M, \tilde{M} \in \mathbb{R}^{d \times d}$ are symmetric matrices characterizing the curvature of the loss landscapes, and $c, \tilde{c} \in \mathbb{R}$ are constants. The parameters $\theta^*, M, c$ of the test loss are unknown, while $\tilde{\theta}^*, \tilde{M}, \tilde{c}$ of the training loss can be estimated from data. Our goal is to understand how the expected test loss landscape relates to the observed training loss landscape and to devise an optimization strategy that minimizes $L(\theta)$ by appropriately manipulating $\tilde{L}(\theta)$.

## 3.2 EXPECTED TEST LOSS LANDSCAPE RESCALES THE TRAINING LOSS LANDSCAPE

We begin by examining the relationship between the expected test loss landscape and the observed training loss landscape under the assumption of quadratic losses. We show that, under certain conditions, the expected test loss can be expressed as a rescaled version of the training loss.

The intuition behind this result is that, while the training loss provides an estimate of the true loss, it is subject to sampling variability and noise. By modeling the loss functions as random quadratics, we can analyze how the expected test loss relates to the observed training loss. Specifically, we aim to determine how the curvature (represented by the Hessian matrices) and the minima of the two loss functions are related in expectation.

**Theorem 1.** *Consider an unknown quadratic loss function:*

$$L(\theta) = \frac{1}{2}(\theta - \theta^*)^T M(\theta - \theta^*) + c \tag{3}$$

*where $\theta^* \in \mathbb{R}^d$, $M \in \mathbb{R}^{d \times d}$ and $c \in \mathbb{R}$ are drawn from a known distribution. Without loss of generality, we assume $M$ is symmetric. Suppose we observe another random quadratic loss $\tilde{L}(\theta)$:*

$$\tilde{L}(\theta) = \frac{1}{2}(\theta - \tilde{\theta}^*)^T \tilde{M}(\theta - \tilde{\theta}^*) + \tilde{c} \tag{4}$$

*where $\tilde{\theta}^*$, $\tilde{M}$ and $\tilde{c}$ are random variables dependent on $\theta^*$, $M$ and $c$. Again, suppose $\tilde{M}$ is symmetric. Assume, $\theta^*, \tilde{\theta}^* \perp M, \tilde{M}, c, \tilde{c}$ and $M \perp \tilde{c}|\tilde{M}$, where $\perp$ indicates independence. Furthermore,*

assume $p_{\theta^*, \tilde{\theta}^*} = p_{\tilde{\theta}^*, \theta^*}$, where $p$ denotes probability density. We also assume the following rotation invariance conditions:

$$p_{M|\tilde{M}}(UMU^T|U\tilde{M}U^T) = p_{M|\tilde{M}}(M|\tilde{M}) \tag{5}$$

for all orthogonal matrices $U$ and $\mathbb{E}[M|\tilde{M}]$ is diagonal when $\tilde{M}$ is diagonal. Suppose for all $\theta$,

$$\mathbb{E}[\tilde{L}(\theta)|\theta^*, M, c] = L(\theta) \tag{6}$$

Denote $\tilde{Q}\tilde{\Lambda}\tilde{Q}^T$ the diagonalization of $\tilde{M}$ for some diagonal matrix $\tilde{\Lambda}$ and orthogonal matrix $\tilde{Q}$. Then,

$$\mathbb{E}[L(\theta)|\tilde{\theta}^*, \tilde{M}, \tilde{c}] = \frac{1}{2}(\theta - \tilde{\theta}^*)^T \tilde{Q} D(\tilde{\Lambda}) \tilde{Q}^T (\theta - \tilde{\theta}^*) + C(\tilde{\theta}^*, \tilde{M}, \tilde{c}) \tag{7}$$

for some function $D$ that outputs a diagonal matrix and function $C$ outputting a scalar.

Please see Appendix A for a proof and Appendix C for justifications of our theoretical assumptions (including the rotational invariance of conditional distributions and independence assumptions). The proof involves leveraging the rotational invariance and the independence assumptions to show that the expected test loss maintains the same eigenvectors as the training loss but with rescaled eigenvalues. This implies that the curvature (Hessian) of the expected test loss is a rescaled version of that of the training loss, aligned along the same principal directions.

Theorem 1 intuitively suggests that the curvature directions of the expected test loss are aligned with those of the training loss but rescaled in each direction. This rescaling affects the sharpness of the loss landscape in different directions, suggesting that optimizing for flatter regions in the training loss (as in SAM algorithms) may not necessarily correspond to optimizing the expected test loss without additional assumptions.

### 3.3 A SERIES OF PERTURBATIONS TRANSFORMS THE TRAINING LOSS TO THE TEST LOSS

Having established the relationship between the expected test loss and the training loss, we now explore how to better approximate the gradient of the test loss using the training loss. We show that a sequence of perturbations applied to the parameters allows us to transform the training loss gradient into an approximation of the test loss gradient.

The key idea is that higher-order derivatives of the training loss can capture information about the curvature of the true loss landscape. By recursively computing these derivatives through perturbations, we can construct a series that approximates the effect of rescaling the eigenvalues in the Hessian of the training loss, effectively transforming it into the Hessian of the test loss.

**Theorem 2.** *Consider two quadratic loss functions:*

$$\tilde{L}(\theta) = \frac{1}{2}(\theta - \tilde{\theta}^*)^T \tilde{M}(\theta - \tilde{\theta}^*) + \tilde{c} \tag{8}$$

$$\bar{L}(\theta) = \frac{1}{2}(\theta - \tilde{\theta}^*)^T \tilde{Q} f(\tilde{\Lambda}) \tilde{Q}^T (\theta - \tilde{\theta}^*) + \bar{c} \tag{9}$$

*where $\tilde{M} \in \mathbb{R}^{d \times d}$ is symmetric with eigendecomposition $\tilde{M} = \tilde{Q}\tilde{\Lambda}\tilde{Q}^T$, and $f$ is an elementwise continuous function satisfying $f(0) = 0, f'(0) = 1$. Suppose elements of $\tilde{\Lambda}$ are bounded. Define $D^t(\theta) \in \mathbb{R}^d$ recursively as:*

$$D^1(\theta) = \nabla \tilde{L}(\theta) \tag{10}$$

$$D^t(\theta) = \frac{\partial}{\partial \zeta} D^1(\theta + \zeta D^{t-1}(\theta))|_{\zeta=0} \tag{11}$$

*for $t > 1$. Then, for all $\epsilon > 0$, there exists a sequence $\gamma_1, \gamma_2, ...\gamma_T \in \mathbb{R}$ such that:*

$$||\nabla \bar{L}(\theta) - \nabla \tilde{L}(\hat{\theta})|| \leq \epsilon ||\theta - \tilde{\theta}^*|| \tag{12}$$

*where*

$$\hat{\theta} = \theta + \sum_{t=1}^{T} \gamma_t D^t(\theta) \tag{13}$$

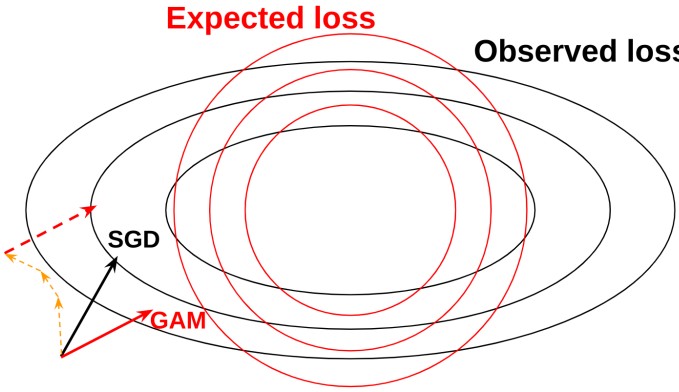

Figure 1: Schematic diagram of generalization aware minimization (GAM) vs stochastic gradient descent (SGD). Black and red contour lines indicate the observed and expected loss landscapes. Note that the expected loss landscape corresponds to the observed loss landscape with rescaled contour lines. SGD takes a gradient step directly against the gradient of the observed loss (black arrow). GAM first makes a series of perturbations in the parameters (orange dashed arrows), computes the update step from the observed loss at the perturbed value (red dashed arrow), and applies the step at the original parameter value (red arrow). This direction corresponds to gradient descent on the expected loss.

Please see Appendix B for a proof. The proof constructs the perturbation coefficients $\gamma_t$ to approximate the effect of the function $f$ on the eigenvalues of $\tilde{M}$. By iteratively applying directional derivatives along the sequence $D^t(\theta)$, we adjust the gradient of the training loss to closely match that of the transformed loss $\bar{L}(\theta)$. See Appendix C for justifications of our theoretical assumptions (including assumptions on the function $f$). Notably, we assume the eigenvalue transformation $f$ is elementwise, differing from the more general eigenvalue transformation derived in Theorem 1; in essence, this assumes orthogonal directions in the parameter space can be treated independently.

Theorem 2 provides a method to approximate the gradient of the expected test loss by applying a series of specific perturbations to the parameters and computing higher-order derivatives of the training loss. This result suggests that we can design an optimization algorithm that adjusts the parameter updates based on these perturbations to directly minimize the expected test loss.

### 3.4 A Practical Online Algorithm to Learn Perturbations

We now extend the theoretical approach to general loss functions by considering local quadratic approximations. We propose a practical algorithm that continually adapts the perturbation sizes over the course of training to improve generalization.

**Algorithm Design**  Based on Theorem 2, we design the *Generalization-Aware Minimization* (GAM) algorithm. GAM uses multiple perturbation steps with higher-order derivatives and tunes the perturbation coefficients $\gamma_t$ online during training. This allows GAM to approximate gradients of the expected test loss instead of using training loss gradients as illustrated in Figure 1. The key components of GAM are:

- **Higher-Order Perturbations:** We compute a sequence of directional derivatives $D^t(\theta)$ to capture higher-order information about the loss landscape.
- **Adaptive Perturbation Sizes:** We update the perturbation coefficients $\gamma_t$ by minimizing a discrepancy function $\Delta$ between the gradient on perturbed parameters and an estimate of the test loss gradient; in practice, we use negative dot product as our discrepancy measure.
- **Auxiliary Minibatches:** We use the training loss on auxiliary minibatches as a proxy for the test loss to guide the adaptation of $\gamma_t$.

Notably, if the number of perturbation steps is fixed at one and $\gamma_1$ is fixed at a constant value, we recover SAM exactly: thus, SAM is a special case of GAM.

**Algorithm Details**    Algorithm 1 outlines the steps of GAM.

---

**Algorithm 1** Generalization Aware Minimization

---

**Require:** Initial parameters $\theta^0$, training set $\mathcal{D}$, GAM steps $T$, training iterations $N$, gradient discrepancy function $\Delta$, small constant $\epsilon > 0$

1: Initialize $\gamma_1, \gamma_2, ...\gamma_T = 0$
2: Initialize $\theta = \theta^0$
3: Sample minibatch $\bar{X}, \bar{Y} \sim \mathcal{D}$
4: **for** iteration $= 1, ..., N$ **do**
5:    Sample minibatch $X, Y \sim \mathcal{D}$
6:    $d^1 = \nabla L(\theta, (X, Y))$
7:    **for** $t = 2, ...T$ **do**
8:       $d^t = \frac{1}{\epsilon}(\nabla L(\theta + \epsilon d^{t-1}, (X, Y)) - d^1)$
9:    **end for**
10:   $\hat{\theta} = \theta + \sum_{t=1}^{T} \gamma_t d^t$
11:   $g_\theta = \nabla L(\hat{\theta}, (X, Y))$
12:   $\bar{g}_\theta = \nabla L(\theta, (\bar{X}, \bar{Y}))$
13:   $g_\gamma = \frac{\partial}{\partial \gamma_1, \gamma_2, ...\gamma_T} \Delta(g_\theta, \bar{g}_\theta)$
14:   Update $\gamma$ following $-g_\gamma$
15:   Update $\theta$ following $-g_\theta$
16: **end for**
17: Return $\theta$

---

**Explanation of Steps**

- **Lines 6–9 (Higher-Order Derivatives):** We recursively compute the directional derivatives $d^t$ using finite differences. The small constant $\epsilon$ ensures numerical stability.

- **Line 10 (Perturbed Parameters):** We obtain the perturbed parameter vector $\hat{\theta}$ by combining the directional derivatives weighted by the coefficients $\gamma_t$.

- **Lines 11–12 (Gradient Computation):** We compute the gradient at the perturbed parameters $g_\theta$ and the gradient on the auxiliary minibatch $\bar{g}_\theta$, which serves as a proxy for the test loss gradient.

- **Line 13 (Perturbation Coefficient Update):** We compute the gradient of the discrepancy between $g_\theta$ and $\bar{g}_\theta$ with respect to the perturbation coefficients and update $\gamma_t$ accordingly.

- **Line 14 (Parameter Update):** We perform a standard gradient descent update on the parameters using $g_\theta$.

**Practical Considerations**

- **Computational Overhead:** Computing higher-order derivatives increases computational cost. However, since we use finite differences and a small number of perturbation steps $T$, the overhead remains manageable.

- **Stability and Convergence:** The choice of $\epsilon$ and the learning rates $\eta_\gamma, \eta_\theta$ can affect the stability of the algorithm. In practice, these hyperparameters are tuned based on validation performance.

- **Extension to Non-Quadratic Losses:** While theoretically motivated for quadratic losses, GAM can be applied to general loss functions by assuming local quadraticity. This allows GAM to be used with complex neural networks and loss functions encountered in deep learning.

**Empirical Validation**    To empirically validate that GAM effectively learns the perturbation coefficients $\gamma_t$, we conduct experiments on a synthetic quadratic optimization problem where the exact relationship between the observed (training) loss and the expected (test) loss is known. In Appendix D, we find that GAM can indeed approximate the correct transformation mapping from the training loss landscape to the expected test loss landscape.

Table 1: Test set accuracies of various network architectures trained on the MNIST, CIFAR-10 and SVHN datasets with different methods: stochastic gradient descent (SGD), sharpness aware minimization (SAM) with different parameter values $\gamma_1$, and generalization aware minimization (GAM). Mean results and standard errors are reported over $5$ trials where applicable. Best results are bolded.

| Method | MNIST | | CIFAR-10 | | | SVHN | | | Imagenet |
|---|---|---|---|---|---|---|---|---|---|
| | 3-layer MLP | 3-layer CNN | 3-layer MLP | 5-layer CNN | 14-layer CNN | 3-layer MLP | 5-layer CNN | 14-layer CNN | ResNet-50 |
| SGD | 0.97368 $\pm$ 0.00072 | 0.96216 $\pm$ 0.00117 | 0.54500 $\pm$ 0.00216 | 0.66334 $\pm$ 0.00104 | 0.83430 $\pm$ 0.00237 | 0.79729 $\pm$ 0.00223 | 0.87824 $\pm$ 0.00055 | 0.93852 $\pm$ 0.00084 | 0.64928 |
| SAM 0.001 | 0.97356 $\pm$ 0.00082 | 0.96224 $\pm$ 0.00111 | 0.54552 $\pm$ 0.00241 | 0.66716 $\pm$ 0.00123 | 0.83138 $\pm$ 0.00231 | 0.79909 $\pm$ 0.00246 | 0.87919 $\pm$ 0.00033 | 0.93937 $\pm$ 0.00069 | - |
| SAM 0.01 | 0.97352 $\pm$ 0.00032 | 0.96244 $\pm$ 0.00117 | 0.54404 $\pm$ 0.00216 | 0.66974 $\pm$ 0.00232 | 0.83328 $\pm$ 0.00213 | 0.80234 $\pm$ 0.00169 | 0.87853 $\pm$ 0.00033 | 0.94156 $\pm$ 0.00048 | - |
| SAM 0.1 | 0.97466 $\pm$ 0.00046 | 0.96298 $\pm$ 0.00088 | 0.55394 $\pm$ 0.0017 | 0.68100 $\pm$ 0.0014 | 0.84288 $\pm$ 0.00345 | 0.80627 $\pm$ 0.0022 | 0.88334 $\pm$ 0.00095 | 0.94252 $\pm$ 0.00143 | 0.65232 |
| CRSAM | 0.97038 $\pm$ 0.00149 | 0.93852 $\pm$ 0.01727 | 0.55402 $\pm$ 0.00148 | **0.69444** $\pm$ **0.00117** | **0.85892** $\pm$ **0.00059** | 0.80181 $\pm$ 0.00221 | **0.89792** $\pm$ **0.00119** | **0.95383** $\pm$ **0.00113** | - |
| GAM | **0.97518** $\pm$ **0.00063** | **0.96392** $\pm$ **0.00036** | **0.56356** $\pm$ **0.00162** | 0.69396 $\pm$ 0.00154 | 0.85074 $\pm$ 0.00122 | **0.81175** $\pm$ **0.00217** | 0.88357 $\pm$ 0.00057 | 0.94299 $\pm$ 0.00025 | **0.65924** |

## 4 RESULTS

### 4.1 EXPERIMENTAL SETUP

In this section, we evaluate the performance of Generalization-Aware Minimization (GAM) in comparison to Sharpness-Aware Minimization (SAM), Curvature-regularized SAM (CRSAM) Wu et al. (2024), and standard stochastic gradient descent (SGD) on standard image classification benchmarks: MNIST (Deng, 2012), CIFAR-10 (Krizhevsky et al., 2009), SVHN (Netzer et al., 2011) and ImageNet (Deng et al., 2009).

For SAM, which corresponds to a special case of GAM with $T = 1$ and fixed perturbation size $\gamma_1$, we experiment with perturbation magnitudes $\gamma_1 \in \{0.001, 0.01, 0.1\}$ to evaluate its sensitivity to this hyperparameter. GAM adaptively learns the perturbation coefficients $\gamma_t$ during training using multiple perturbation steps ($T > 1$) and higher-order gradient information.

All methods are trained using the same optimization settings, including learning rates, batch sizes, and training epochs, to ensure a fair comparison. Detailed architectures, hyperparameter settings and training procedures are provided in Appendix E. Appendix F Figure 6 shows the performance of GAM under different hyperparameter settings; in summary, we find that GAM can become unstable under large $T$, performs best at small batch sizes, and is relatively insensitive to $\epsilon$.

### 4.2 GAM OUTPERFORMS SAM ON BENCHMARKS

Table 1 presents the test accuracies achieved by each method across different datasets and network architectures. The results demonstrate that GAM consistently and statistically significantly outperforms both SAM and standard SGD on all benchmarks. For instance, for ResNet–50 trained on ImageNet, GAM achieves a test accuracy of **65.92%**, surpassing SAM's performance of 65.23%. Nevertheless, on certain architectures, CRSAM outperforms GAM by a notable margin indicating GAM may not be universally better than all SAM variants.

Appendix F Figure 5 shows the test error over the course of training for each method. Notably, GAM may underperform relative to other methods during the early stages of training as it learns the optimal perturbation coefficients $\gamma_t$. However, as training progresses, GAM adjusts these coefficients effectively, leading to superior generalization performance by the end of training. This adaptive learning of perturbations allows GAM to fine-tune its optimization strategy based on the evolving loss landscape. These results suggest that GAM's ability to adaptively learn perturbation sizes and use higher-order gradient information contributes to its enhanced generalization performance across different models and datasets.

### 4.3 ANALYZING THE TRANSFORMATION FROM TRAINING TO TEST LOSS

To gain further insight into how GAM improves generalization, we analyze the transformation applied by GAM to the loss landscape. Specifically, we examine how GAM modifies the eigenvalues

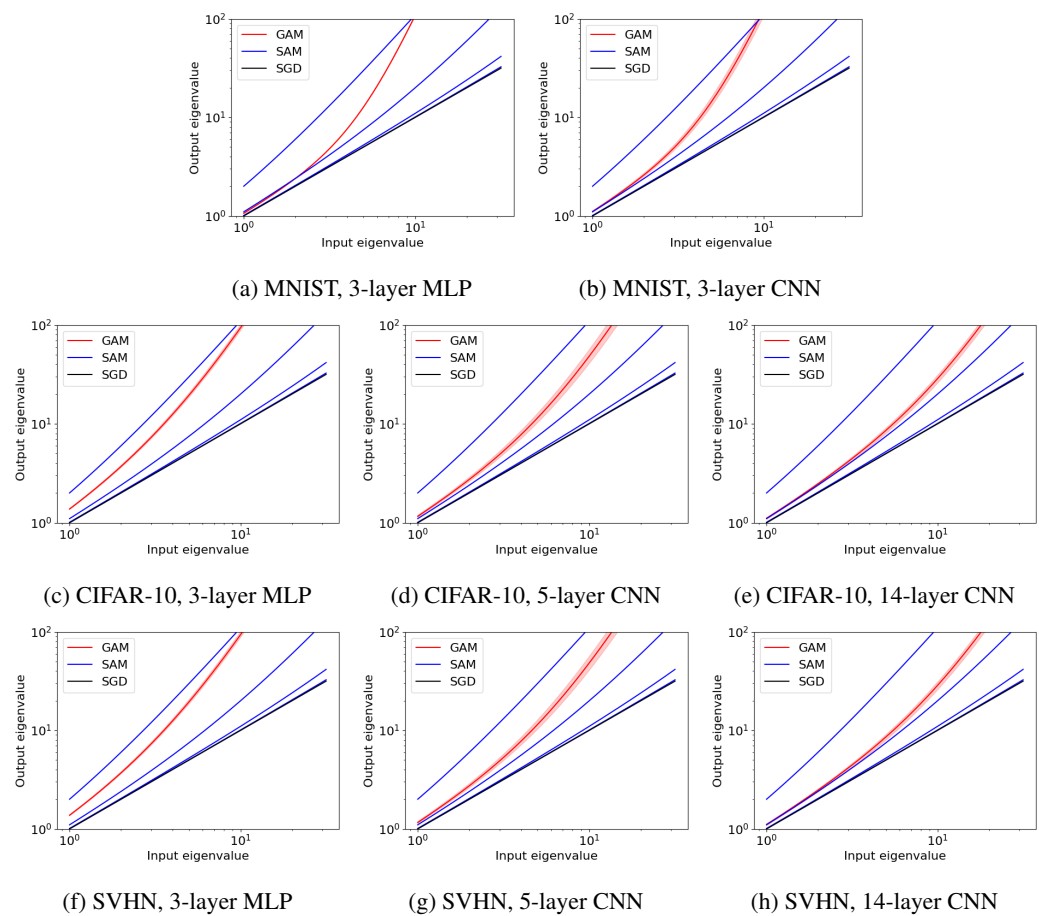

(a) MNIST, 3-layer MLP      (b) MNIST, 3-layer CNN

(c) CIFAR-10, 3-layer MLP    (d) CIFAR-10, 5-layer CNN    (e) CIFAR-10, 14-layer CNN

(f) SVHN, 3-layer MLP      (g) SVHN, 5-layer CNN      (h) SVHN, 14-layer CNN

Figure 2: Visualization for different dataset-architecture combinations of Hessian eigenvalue transformation of different training methods: stochastic gradient descent (SGD), sharpness aware minimization (SAM) with different parameter values $\gamma_1$, and generalization aware minimization (GAM). Input eigenvalue corresponds to the observed loss Hessian while the output eigenvalue corresponds to the transformed loss Hessian. Margins for GAM indicate standard errors over 5 trials.

of the Hessian of the training loss, which correspond to the curvature along different parameter directions. Figure 2 visualizes the relationship between the original Hessian eigenvalues (input eigenvalues) and the transformed eigenvalues (output eigenvalues) for different methods. For GAM, we compute the effective transformation induced by the learned perturbation coefficients $\gamma_t$. We observe that GAM tends to sharpen already sharp directions (i.e., directions with large eigenvalues), qualitatively similar to SAM. As GAM is tuned specifically to optimize for generalization, the results suggest that SAM generalizes well because of its similarity to GAM. Interestingly, GAM's transformation exhibits a higher contrast between small and large curvature directions than SAM, selectively sharpening sharp directions while maintaining others. This more complex behavior arises from the higher-order gradient information used by GAM, ultimately yielding better generalization.

## 4.4 MITIGATING GAM'S COMPUTATIONAL COST

One of GAM's disadvantages is its computational cost relative to SGD and SAM: it requires computing the derivative of parameter updates with respect to perturbation coefficients $\gamma_t$ which can be quite costly. We propose mitigating this cost by updating the $\gamma_t$ periodically instead of at each training step as done in Algorithm 1. As shown in Figure 3, when updating the $\gamma_t$ every time step, GAM's computational cost is roughly $4\times$ that of SGD (relative to roughly $1.3\times$ for SAM). However, this cost can be reduced to roughly $3\times$ when updating the perturbation coefficients periodically. Although this reduces test accuracy, the accuracy of GAM still exceeds that of SAM or SGD.

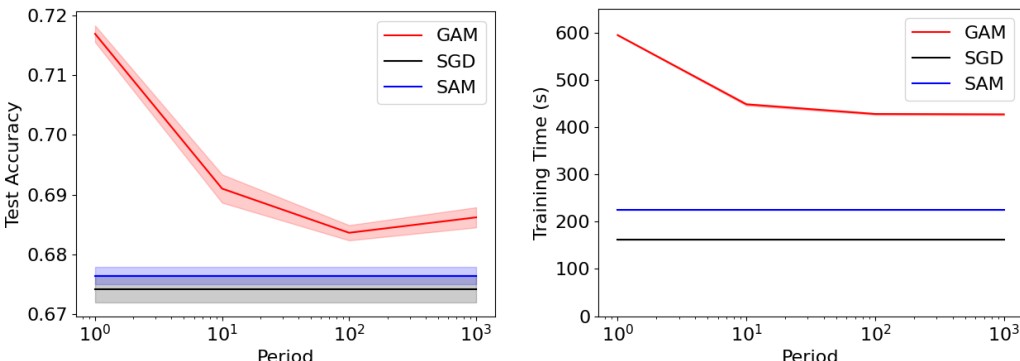

Figure 3: Test set accuracies and training time of a 5-layer CNN model trained on the CIFAR-10 dataset with stochastic gradient descent (SGD), sharpness aware minimization (SAM) with $\gamma_1 = 0.01$, and generalization aware minimization (GAM). The x-axis indicates the period at which GAM updates its perturbation coefficients. Mean results and standard errors are reported over 5 trials. Training times are on an NVIDIA RTX 4090 GPU.

## 5 DISCUSSION

In this work, we introduced Generalization-Aware Minimization (GAM), a novel optimization algorithm that directly targets the expected test loss by using higher-order gradient information and adaptive perturbations. Unlike Sharpness-Aware Minimization (SAM) algorithms, which rely on the heuristic that flatter regions of the loss landscape generalize better, GAM is grounded in a theoretical framework that aligns the optimization process with the expected test loss. By demonstrating that the expected test loss landscape is a rescaled version of the observed training loss landscape for quadratic losses, we provided a principled approach to improve generalization.

The surprising similarity between the update mechanisms of SAM and GAM offers a new perspective on why SAM improves generalization. Our analysis suggests that SAM may implicitly approximate the expected test loss through its single-step perturbations, which could explain its empirical success. However, GAM's use of multiple perturbation steps and higher-order derivatives allows it to more accurately capture the transformation between the training and test loss landscapes. Our empirical results on benchmark datasets confirm that GAM consistently outperforms SAM, highlighting the benefits of directly optimizing for generalization.

GAM shows promising results in real settings, however it relies on using higher-order derivatives, which may be computationally challenging for large or non-differentiable networks. Future work could explore approximations or scalable implementations of higher-order derivatives. We also note the possibility of integrating GAM with newer variants of SAM such as CRSAM which in some cases can outperform GAM.

We believe that GAM opens new avenues for developing optimization algorithms that can further enhance generalization in deep learning models, potentially leading to more robust and reliable AI systems. By incorporating higher-order gradient information and adaptive strategies, future optimizers can more effectively navigate the loss landscape to find solutions that generalize well. We hope that our work inspires further research into optimization techniques that bridge theoretical insights and practical performance, ultimately contributing to the advancement of generalization in machine learning.

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

## A    PROOF OF THEOREM 1

*Proof.* First, observe that

$$\mathbb{E}[L(\theta)|\tilde{\theta}^*, \tilde{M}, \tilde{c}] = \mathbb{E}[\frac{1}{2}(\theta - \theta^*)^T M(\theta - \theta^*) + c|\tilde{\theta}^*, \tilde{M}, \tilde{c}]$$

$$= \frac{1}{2}\theta^T \mathbb{E}[M|\tilde{\theta}^*, \tilde{M}, \tilde{c}]\theta - \theta^T \mathbb{E}[M\theta^*|\tilde{\theta}^*, \tilde{M}, \tilde{c}] + \mathbb{E}[c|\tilde{\theta}^*, \tilde{M}, \tilde{c}] \quad (14)$$

Since $\theta^*$ and $\tilde{\theta}^*$ are independent of $\tilde{M}$ and $\tilde{c}$, and $M$ is independent of $\tilde{\theta}^*$, we have:

$$= \frac{1}{2}\theta^T \mathbb{E}[M|\tilde{M}, \tilde{c}]\theta - \theta^T \mathbb{E}[M|\tilde{M}, \tilde{c}]\mathbb{E}[\theta^*|\tilde{\theta}^*] + \mathbb{E}[c|\tilde{\theta}^*, \tilde{M}, \tilde{c}] \quad (15)$$

Factoring:

$$= \frac{1}{2}(\theta - \mathbb{E}[\theta^*|\tilde{\theta}^*])^T \mathbb{E}[M|\tilde{M}, \tilde{c}](\theta - \mathbb{E}[\theta^*|\tilde{\theta}^*]) + \frac{1}{2}\mathbb{E}[\theta^*|\tilde{\theta}^*]^T \mathbb{E}[M|\tilde{M}, \tilde{c}]\mathbb{E}[\theta^*|\tilde{\theta}^*] + \mathbb{E}[c|\tilde{\theta}^*, \tilde{M}, \tilde{c}]$$
$$(16)$$

Letting $C(\tilde{\theta}^*, \tilde{M}, \tilde{c}) = \frac{1}{2}\mathbb{E}[\theta^*|\tilde{\theta}^*]^T \mathbb{E}[M|\tilde{M}, \tilde{c}]\mathbb{E}[\theta^*|\tilde{\theta}^*] + \mathbb{E}[c|\tilde{\theta}^*, \tilde{M}, \tilde{c}]$:

$$\mathbb{E}[L(\theta)|\tilde{\theta}^*, \tilde{M}, \tilde{c}] = \frac{1}{2}(\theta - \mathbb{E}[\theta^*|\tilde{\theta}^*])^T \mathbb{E}[M|\tilde{M}, \tilde{c}](\theta - \mathbb{E}[\theta^*|\tilde{\theta}^*]) + C(\tilde{\theta}^*, \tilde{M}, \tilde{c}) \quad (17)$$

Next, we use the fact that $\mathbb{E}[\tilde{L}(\theta)|\theta^*, M, c] = L(\theta)$. Expanding using the definition of $\tilde{L}(\theta)$ and $L(\theta)$:

$$\frac{1}{2}(\theta - \theta^*)^T M(\theta - \theta^*) + c = \frac{1}{2}\theta^T \mathbb{E}[\tilde{M}|\theta^*, M, c]\theta - \theta^T \mathbb{E}[\tilde{M}\tilde{\theta}^*|\theta^*, M, c] + \mathbb{E}[\tilde{c}|\theta^*, M, c] \quad (18)$$

Once again using the independence between $\theta^*$ and $\tilde{\theta}^*$ from $M$ and $c$, and the independence of $\tilde{M}$ is $\theta^*$:

$$\frac{1}{2}(\theta - \theta^*)^T M(\theta - \theta^*) + c = \frac{1}{2}\theta^T \mathbb{E}[\tilde{M}|M, c]\theta - \theta^T \mathbb{E}[\tilde{M}|M, c]\mathbb{E}[\tilde{\theta}^*|\theta^*] + \mathbb{E}[\tilde{c}|\theta^*, M, c] \quad (19)$$

Since this holds for all $\theta$, we may equate coefficients:

$$M = \mathbb{E}[\tilde{M}|M, c] \quad (20)$$

$$\theta^* = \mathbb{E}[\tilde{\theta}^*|\theta^*] \quad (21)$$

Next, note that $p_{\theta^*, \tilde\theta^*} = p_{\tilde\theta^*, \theta^*}$ implies that $\theta^*$ and $\tilde\theta^*$ have the same marginal distributions, and same conditional distributions conditioned on each other. Since $\theta^* = \mathbb{E}[\tilde\theta^* | \theta^*]$, by symmetry, we must have:

$$\tilde\theta^* = \mathbb{E}[\theta^* | \tilde\theta^*] \tag{22}$$

Thus, we may write the expectation of $L(\theta)$ as:

$$\mathbb{E}[L(\theta) | \tilde\theta^*, \tilde M, \tilde c] = \frac{1}{2}(\theta - \tilde\theta^*)^T \mathbb{E}[M | \tilde M, \tilde c](\theta - \tilde\theta^*) + C(\tilde\theta^*, \tilde M, \tilde c) \tag{23}$$

Next, we consider $\mathbb{E}[M | \tilde M, \tilde c]$. Since $M \perp \tilde c | \tilde M$, we have $\mathbb{E}[M | \tilde M, \tilde c] = \mathbb{E}[M | \tilde M]$. Expanding:

$$\mathbb{E}[M | \tilde M, \tilde c] = \sum_M M p_{M | \tilde M}(M | \tilde M) \tag{24}$$

We denote the eigendecomposition of $\tilde M = \tilde Q \tilde\Lambda \tilde Q^T$. Note that since $\tilde M$ is symmetric, $\tilde Q$ is orthogonal. Substituting:

$$\mathbb{E}[M | \tilde M, \tilde c] = \sum_M M p_{M | \tilde M}(M | \tilde Q \tilde\Lambda \tilde Q^T) \tag{25}$$

By the rotation invariance of $p_{M | \tilde M}$, we have:

$$\mathbb{E}[M | \tilde M, \tilde c] = \sum_M M p_{M | \tilde M}(\tilde Q^T M \tilde Q | \tilde\Lambda) \tag{26}$$

Making a change of variables in the summation, $M' = \tilde Q^T M \tilde Q$:

$$\mathbb{E}[M | \tilde M, \tilde c] = \tilde Q [\sum_{M'} M' p_{M | \tilde M}(M' | \tilde\Lambda)] \tilde Q^T \tag{27}$$

Note that the term in the brackets is simply $\mathbb{E}[M | \tilde\Lambda]$ which is diagonal by assumption. Thus, for some diagonal matrix $D(\tilde\Lambda)$, we may write

$$\mathbb{E}[M | \tilde M, \tilde c] = \tilde Q D(\tilde\Lambda) \tilde Q^T \tag{28}$$

Finally, the expectation of $L(\theta)$ becomes:

$$\mathbb{E}[L(\theta) | \tilde\theta^*, \tilde M, \tilde c] = \frac{1}{2}(\theta - \tilde\theta^*)^T \tilde Q D(\tilde\Lambda) \tilde Q^T (\theta - \tilde\theta^*) + C(\tilde\theta^*, \tilde M, \tilde c) \tag{29}$$

$\square$

## B    PROOF OF THEOREM 2

*Proof.* First, observe that

$$D^t(\theta) = \tilde M^t(\theta - \tilde\theta^*) \tag{30}$$

We may see this by induction. $D^1(\theta) = \nabla \tilde L(\theta) = \tilde M(\theta - \tilde\theta^*)$. If $D^t(\theta) = \tilde M^t(\theta - \tilde\theta^*)$, then

$$D^{t+1}(\theta) = \frac{\partial}{\partial \zeta} D^1(\theta + \zeta D^t(\theta))|_{\zeta=0} = \frac{\partial}{\partial \zeta} \tilde M[\theta + \zeta \tilde M^t(\theta - \tilde\theta^*) - \tilde\theta^*]|_{\zeta=0}$$

$$= \frac{\partial}{\partial \zeta} \tilde M(\theta - \tilde\theta^*) + \zeta \tilde M^{t+1}(\theta - \tilde\theta^*)|_{\zeta=0} = M^{t+1}(\theta - \tilde\theta^*) \tag{31}$$

Now consider $\nabla \bar L(\theta)$ and $\nabla \tilde L(\hat\theta)$.

$$\nabla \bar L(\theta) = \tilde Q f(\tilde\Lambda) \tilde Q^T (\theta - \tilde\theta^*) \tag{32}$$

and

$$\nabla \tilde L(\hat\theta) = \tilde M(\hat\theta - \tilde\theta^*) = \tilde M(\theta - \tilde\theta^* + \sum_{t=1}^{T} \gamma_t \tilde M^t(\theta - \tilde\theta^*)) = (\tilde M + \sum_{t=1}^{T} \gamma_t \tilde M^{t+1})(\theta - \tilde\theta^*) \tag{33}$$

Using the eigendecomposition of $\tilde{M}$, we have:

$$\nabla \tilde{L}(\hat{\theta}) = \tilde{Q}(\tilde{\Lambda} + \sum_{t=1}^{T} \gamma_t \tilde{\Lambda}^{t+1}) \tilde{Q}^T (\theta - \tilde{\theta}^*) \tag{34}$$

Now, we compare $f(\tilde{\Lambda})$ to $\tilde{\Lambda} + \sum_{t=1}^{T} \gamma_t \tilde{\Lambda}^{t+1}$. Observe that the function $P(\tilde{\lambda}) = \tilde{\lambda} + \sum_{t=1}^{T} \gamma_t \tilde{\lambda}^{t+1}$ can represent any polynomial with intercept $P(0) = 0$ and slope $P'(0) = 1$. By the Weierstrass approximation theorem (Weierstrass, 1885), because the elements of $\tilde{\Lambda}$ are bounded and $f$ is continuous, we may construct the following uniform bound:

$$||\tilde{\Lambda} + \sum_{t=1}^{T} \gamma_t \tilde{\Lambda}^{t+1} - f(\tilde{\Lambda})||_F \leq \epsilon \tag{35}$$

for all $\epsilon > 0$ and diagonal $\Lambda$, for some choice of sequence $\gamma_1, \gamma_2, ... \gamma_T$. By the rotation invariance of the Frobenius norm, we have:

$$||\tilde{Q}(\tilde{\Lambda} + \sum_{t=1}^{T} \gamma_t \tilde{\Lambda}^{t+1})\tilde{Q}^T - \tilde{Q}f(\tilde{\Lambda})\tilde{Q}^T||_F \leq \epsilon \tag{36}$$

Finally, since the Frobenius norm is an upper bound on the maximum eigenvalue of a matrix, we have:

$$||\tilde{Q}(\tilde{\Lambda} + \sum_{t=1}^{T} \gamma_t \tilde{\Lambda}^{t+1})\tilde{Q}^T(\theta - \tilde{\theta}^*) - \tilde{Q}f(\tilde{\Lambda})\tilde{Q}^T(\theta - \tilde{\theta}^*)|| = ||\nabla\tilde{L}(\hat{\theta}) - \nabla\bar{L}(\theta)|| \leq \epsilon||\theta - \tilde{\theta}^*|| \tag{37}$$

$\square$

## C  JUSTIFICATION OF THEORETICAL ASSUMPTIONS

In this section, we provide practical justifications for the theoretical assumptions made in Theorems 1 and 2. These assumptions are critical for the validity of our theoretical results and are grounded in common practices and observations in machine learning.

**Assumption 1: $M$ and $\tilde{M}$ are symmetric matrices.**  **Explanation:** In the context of quadratic loss functions, $M$ and $\tilde{M}$ represent the Hessian matrices (second derivatives) of the true loss $L(\theta)$ and the training loss $\tilde{L}(\theta)$, respectively. By definition, Hessian matrices of scalar-valued functions are symmetric because mixed partial derivatives commute (i.e., $\frac{\partial^2 L}{\partial \theta_i \partial \theta_j} = \frac{\partial^2 L}{\partial \theta_j \partial \theta_i}$) when the function is twice continuously differentiable. In practice, loss functions used in machine learning, such as mean squared error and cross-entropy loss, satisfy these smoothness conditions. Therefore, assuming that $M$ and $\tilde{M}$ are symmetric is both standard and justifiable.

**Assumption 2: $\theta^*$ and $\tilde{\theta}^*$ are independent of $M$, $\tilde{M}$, $c$, and $\tilde{c}$.**  **Explanation:** This assumption simplifies the analysis by decoupling the location of the minima from the curvature and offset of the loss functions. In practical terms, it means that the position of the minimum (i.e., the parameter values that minimize the loss) does not influence the curvature of the loss landscape or the constant term. This is a reasonable approximation when considering local behavior around $\theta$, especially in high-dimensional parameter spaces where the curvature is determined by the structure of the model and the data distribution rather than the specific parameter values.

**Assumption 3: $M$ is independent of $\tilde{c}$ given $\tilde{M}$, i.e., $M \perp \tilde{c} \mid \tilde{M}$.**  **Explanation:** This assumption asserts that, conditioned on the training loss curvature $\tilde{M}$, the curvature of the true loss $M$ is independent of the constant offset $\tilde{c}$ of the training loss. In practical scenarios, the constant term $\tilde{c}$ does not affect the gradient or Hessian of the loss function and, therefore, does not influence the optimization process. Since $\tilde{c}$ merely shifts the loss landscape vertically without changing its shape or curvature, it is reasonable to consider $M$ independent of $\tilde{c}$ given $\tilde{M}$.

**Assumption 4: The joint distribution of $\theta^*$ and $\tilde{\theta}^*$ is symmetric, i.e., $p_{\theta^*,\tilde{\theta}^*} = p_{\tilde{\theta}^*,\theta^*}$. Explanation:** This symmetry assumption implies that the statistical relationship between the true minimum $\theta^*$ and the observed training minimum $\tilde{\theta}^*$ is bidirectional and unbiased. In practical terms, it suggests that there is no preferential direction in the estimation errors between $\theta^*$ and $\tilde{\theta}^*$. This is a reasonable assumption when the training data is a representative sample of the underlying data distribution, and there are no systematic biases affecting the estimation of the minima. It facilitates the theoretical analysis by ensuring consistent behavior regardless of the direction of estimation.

**Assumption 5: Rotation invariance condition:** $p_{M|\tilde{M}}(UMU^T|U\tilde{M}U^T) = p_{M|\tilde{M}}(M|\tilde{M})$ **for all orthogonal matrices $U$. Explanation:** The rotation invariance assumption states that the conditional distribution of the true loss curvature $M$ given the training loss curvature $\tilde{M}$ is invariant under orthogonal transformations (rotations) of the parameter space. Practically, this means that the orientation of the coordinate system does not affect the statistical relationship between $M$ and $\tilde{M}$. This assumption is justified in many machine learning models where the parameter space does not have a natural orientation, especially in isotropic settings where all directions are treated equally. It allows us to generalize results without loss of generality and simplifies the analysis by enabling diagonalization of matrices through rotations.

**Assumption 6: The conditional expectation $\mathbb{E}[M|\tilde{M}]$ is diagonal when $\tilde{M}$ is diagonal. Explanation:** This assumption suggests that if the training loss curvature matrix $\tilde{M}$ is diagonal (indicating no interaction between different parameters), then the expected test loss curvature matrix $M$ conditioned on $\tilde{M}$ is also diagonal. In practical terms, when the training loss landscape exhibits axis-aligned curvature, it is reasonable to expect that the true loss landscape will have similar properties in expectation. Without this assumption, we must break the symmetry provided by the training loss landscape by assuming the expected test loss has a different and arbitrary set of curvature axes.

**Assumption 7: For all $\theta$, the expected training loss equals the true loss, i.e., $\mathbb{E}[\tilde{L}(\theta)|\theta^*, M, c] = L(\theta)$. Explanation:** This assumption embodies the idea that, conditioned on the true loss parameters, the training loss is an unbiased estimator of the true loss at any point $\theta$. Practically, this means that the training data provides an accurate reflection of the true loss landscape on average. This assumption is justified when the training data is an independent and identically distributed (i.i.d.) sample from the same distribution as the test data, and there are no systematic errors or biases in the data collection process. It underpins the validity of using the training loss to make inferences about the true loss.

**Assumption 8: The function $f$ is element-wise. Explanation:** In Theorem 1, recall that the eigenvalues of the observed Hessian and the expected test loss Hessian are related by an arbitrary function $D(\tilde{\Lambda})$. Here, we make the assumption that the function is applied *independently* to each eigenvalue. We believe this is reasonable because in many machine learning models, especially those with large numbers of parameters, the interactions between different parameters can often be approximated as negligible. This means that curvature transformation between train and test landscapes for one principle parameter direction is independent of the transformation for another principle parameter direction.

**Assumption 9: The function $f$ is element-wise continuous with $f(0) = 0$ and $f'(0) = 1$. Explanation:** In Theorem 2, $f$ is used to modify the eigenvalues of the training loss curvature matrix $\tilde{M}$ to approximate the curvature of the true loss. The conditions $f(0) = 0$ and $f'(0) = 1$ ensure that $f$ behaves smoothly near zero and that small eigenvalues are not disproportionately affected, which is important for stability. In practice, we believe it is reasonable to expect that very flat directions of the expected test loss correspond to similarly flat directions of the training loss and vice versa, which is what these conditions on $f$ imply.

**Assumption 10: The elements of $\tilde{\Lambda}$ are bounded. Explanation:** The boundedness of the eigenvalues in $\tilde{\Lambda}$ prevents extreme curvature in the training loss landscape. In practice, deep learning models are initialized with weights near $0$ and activation functions with bounded derivatives; thus, it is reasonable to expect curvature to be practically boundable.

## D    EMPIRICAL VALIDATION OF GAM

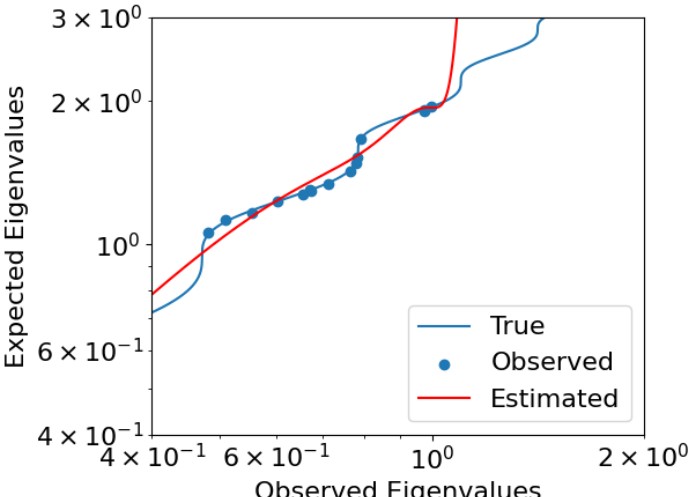

Figure 4: True versus inferred eigenvalue transformation: Red line: Estimated transformation of eigenvalues from training to true loss landscape. Blue line: Actual Hessian eigenvalue transformation. Blue dots: eigenvalues in the observed loss.

To empirically validate that GAM effectively learns the perturbation coefficients $\gamma_t$, we conduct experiments on a synthetic quadratic optimization problem where we control the relationship between the observed (training) loss and the expected (test) loss. This setup allows us to directly assess whether GAM can learn the true transformation from the training loss landscape to the expected test loss landscape.

We consider observed and true quadratic loss landscapes with the same Hessian eigenvectors, but different Hessian eigenvalues. To simulate the observed training Hessian $\tilde{M}$, we define a nonlinear transformation $f$ that relates the observed eigenvalues $\tilde{\lambda}$ to the true eigenvalues $\lambda$:

$$\tilde{\lambda} = f(\lambda) = \frac{1}{2}\lambda + \frac{1}{20}\sin(10\lambda). \tag{38}$$

This transformation introduces both scaling and oscillatory behavior, mimicking complex discrepancies between the training and test loss landscapes. Our objective is to learn the perturbation coefficients $\gamma_t$ such that the gradient of the perturbed training loss $\nabla\tilde{L}(\hat{\theta})$ closely approximates the gradient of the true loss $\nabla L(\theta)$; we use squared error as our discrepancy metric $\Delta$. See Appendix E for further details.

As shown in Figure 4, the estimated transformation approximates the true transformation across the range of Hessian eigenvalues in the observed loss. This indicates that the learned perturbation coefficients effectively capture the nonlinear mapping between the training and test loss landscapes. We highlight, however, that the transformation may be inaccurate outside of the range of observed Hessian eigenvalues.

## E    EXPERIMENTAL DETAILS

All experiments were run on a 24 GB GPU.

### E.1 Synthetic Quadratic Problem

We set the parameter dimension to $d = 15$ and use $T = 12$ perturbation steps. We use a finite difference constant of $\epsilon = 0.1$ to approximate higher order derivatives. The perturbation coefficients $\gamma_t$ are optimized using the Adam (Kingma, 2015) optimizer with a learning rate of $10^{-3}$ over $100000$ training iterations.

We generate the true Hessian $M$ by sampling:

- A random orthogonal matrix $Q \in \mathbb{R}^{d \times d}$ via QR decomposition of a random Gaussian matrix.
- True eigenvalues $\lambda \in \mathbb{R}^d$ sampled uniformly from $[1, 2]$, ensuring positive definiteness.

The true Hessian is then constructed as $M = Q \operatorname{diag}(\lambda) Q^T$.

To simulate noise in the observed Hessian (as would occur due to sampling variability in real datasets), we add Gaussian noise to the true eigenvalues:

$$\lambda_{\text{noisy}} = \lambda + \sigma \cdot \eta, \tag{39}$$

where $\eta \sim \mathcal{N}(0, I)$ and $\sigma = 0.01$. The observed eigenvalues are then computed as $\tilde{\lambda} = f(\lambda_{\text{noisy}})$ using the transformation in Equation 38.

### E.2 MNIST

We consider two networks, 1) a softplus-activated MLP network with 3 fully-connected layers of hidden layer size $256$, 2) a softplus-activated CNN with 2 stride-2, kernel-3 convolutional layers with channel sizes $32$ and $64$, followed by global average pooling and a final linear layer. Each learnable weight layer is preceeded by batch normalization.

We train all methods for 10 epochs with batch size 100 using Adam optimizer (Kingma, 2015) at learning rate $10^{-3}$. For GAM, we use $T = 3$ perturbation steps and tune $\gamma$s using Adam at learning rate $10^{-3}$. All experiments are conducted over 5 random seeds. For GAM, we use the following discrepancy function: $\Delta(g_\theta, \bar{g}_\theta) = -g_\theta^T \bar{g}_\theta$ and set $\epsilon = 10^{-3}$. For CRSAM, we set step size to $0.1$, $\alpha = 0.1$, $\beta = 0.01$.

MNIST is made available to us via a Creative Commons license.

### E.3 CIFAR-10 and SVHN

We consider three networks, 1) a softplus-activated MLP network with 3 fully-connected layers of hidden layer size $1024$, 2) a softplus-activated CNN with 4 stride-2, kernel-3 convolutional layers with channel sizes $32, 64, 128$ and $256$, followed by global average pooling and a final linear layer, 3) a softplus-activated convolutional neural network with 13 convolutional layers followed by global average pooling and a final linear layer. In the 14-layer CNN, all convolutions have stride 1 except for the sixth and tenth, and have channel sizes $16, 32, 32, 32, 32, 64, 64, 64, 64, 128, 128, 128, 128$. Each learnable weight layer is preceeded by batch normalization.

We train all methods for 20 epochs with batch size 100 using Adam optimizer (Kingma, 2015) at learning rate $10^{-3}$. For GAM, we use $T = 2$ perturbation steps and tune $\gamma$s using Adam at learning rate $10^{-3}$. All experiments are conducted over 5 random seeds. For GAM, we use the following discrepancy function: $\Delta(g_\theta, \bar{g}_\theta) = -g_\theta^T \bar{g}_\theta$ and set $\epsilon = 10^{-3}$. For CRSAM, we set step size to $0.1$, $\alpha = 0.1$, $\beta = 0.01$.

SVHN is made available to us for non-commercial use only.

### E.4 ImageNet

We consider a softplus-activated ResNet-50 He et al. (2016) with standard settings; the only replacement is ReLU with softplus.

We train all methods for 20 epochs with batch size 32 using Adam optimizer (Kingma, 2015) at learning rate $10^{-3}$. For GAM, we use $T = 2$ perturbation steps and tune $\gamma$s using Adam at learning rate $10^{-3}$. Due to computational constraints, we run one experimental trial. For GAM, we use the following discrepancy function: $\Delta(g_\theta, \bar{g}_\theta) = -g_\theta^T \bar{g}_\theta$ and set $\epsilon = 10^{-3}$.

ImageNet is made available to us for non-commercial use only.

## F  ADDITIONAL RESULTS

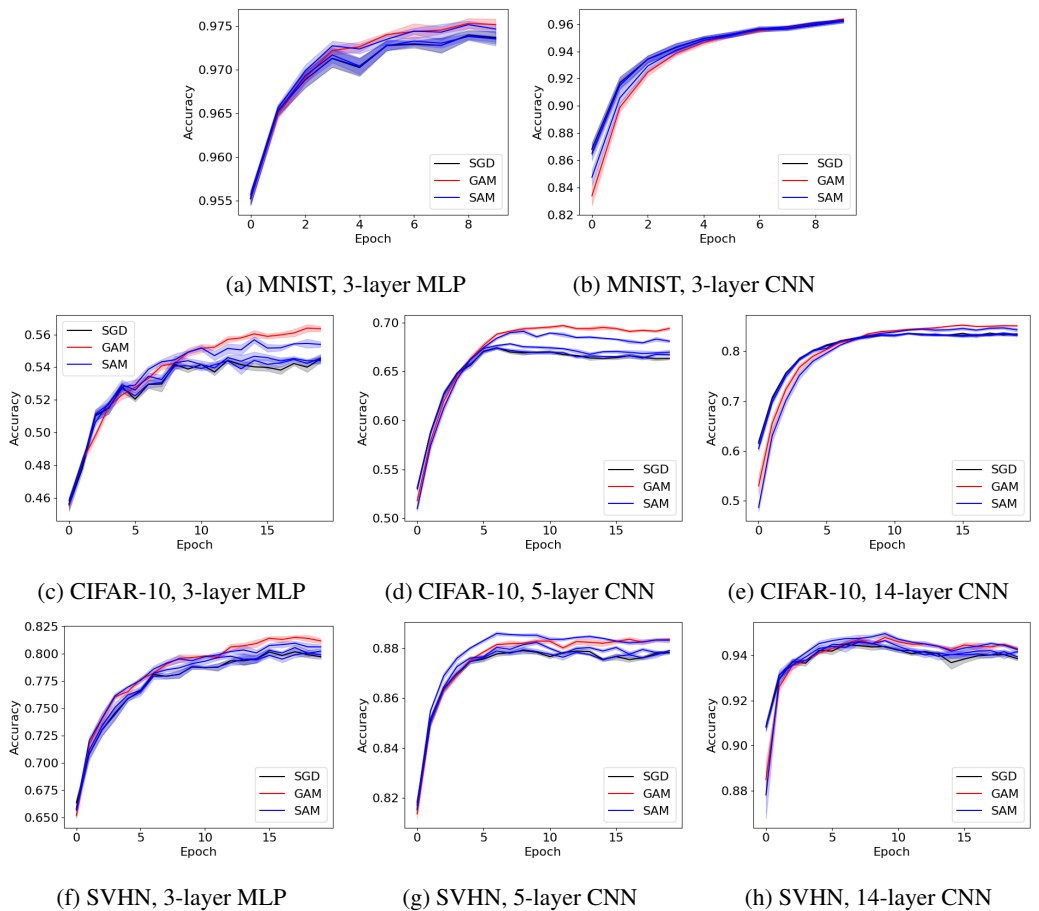

(a) MNIST, 3-layer MLP      (b) MNIST, 3-layer CNN

(c) CIFAR-10, 3-layer MLP    (d) CIFAR-10, 5-layer CNN    (e) CIFAR-10, 14-layer CNN

(f) SVHN, 3-layer MLP      (g) SVHN, 5-layer CNN      (h) SVHN, 14-layer CNN

Figure 5: Test error over the course of training for various networks trained on MNIST, CIFAR-10 and SVHN with different methods: stochastic gradient descent (SGD), sharpness aware minimization (SAM) with different parameter values $\gamma_1$, and generalization aware minimization (GAM). Margins indicate standard errors over 5 trials.

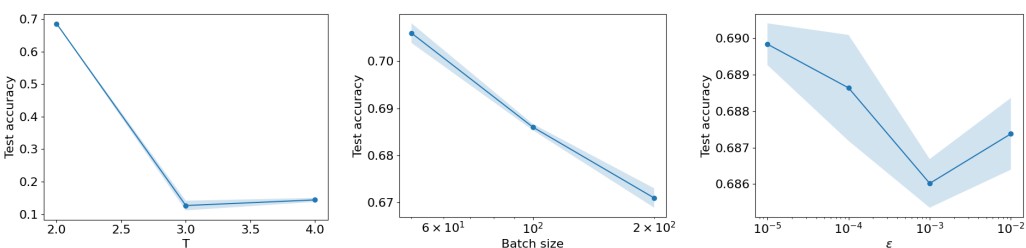

Figure 6: Test set accuracies of a CNN model trained on the CIFAR-10 dataset with generalization aware minimization (GAM) under different hyperparameter choices. By default, we use $T = 2$, batch size of $100$, and $\epsilon = 10^{-3}$. Mean results and standard errors are reported over 5 trials.

