# OpenReview forum: "Generalization Aware Minimization"
_ICLR.cc/2026/Conference — Submitted to ICLR 2026_

### Official Review · Reviewer_NWzY · 2025-10-19

**Soundness:** 2
**Presentation:** 2
**Contribution:** 3
**Rating:** 2
**Confidence:** 4

**Summary:**

This paper developed a new optimizer, GAM, that directly optimizes the expected test loss. This is   different from SAM, which minimizes the so-called flatness instead. The authors argue that the test loss landscape is a rescaled version of the training loss landscape, and GAM is able to approximate the corresponding gradient after a sequence of perturbative updates. The effectiveness of GAM is finally evaluated on several benchmarks.

**Strengths:**

This paper is easy to follow and the flow of the proposed algorithm is very clear. The idea of transforming the gradient of training loss to the gradient of test loss is really impressive, which I believe deserves further study.

**Weaknesses:**

While the proposed algorithm is quite impressive, there remains several issues that needs to addressed:

 - My biggest concern is about the quadratic assumption. In most cases, as far as I can tell, this assumption does not hold during the training process. I suggest the authors could provide more experimental validations.
 - In Theorem 2, the authors showed that the transformation of one gradient to another. Can the authors provide more intepretation on why requiring $f^\prime(0)=1$? Moreover, one can notice from Figure 4 that the transformation seems to be not as expected. This could be more severe for high-dimensional distributions, particularly for millions of parameters. What the authors have done to alleviate this issue?
 - In Algorithm 1, how line 13, the gradient with respect to $\gamma$ is implemented?
 - More experimental results should be included. In Table 1, the authors only reported the results on several simple architectures. And the accuracy on ImageNet-1k is too low (practically, it should be at least 70\% even for SGD). And the legend of Figure 2 should be better indicated such as including the value of $\gamma_1$ and using different styles. In brief, more experiments including baselines, computation overhead, hyper-parameter sensitivy should be also be presented.

**Questions:**

please see Weaknesses.

---

> ### Author Response · Authors · 2025-11-19
>
> Thank you for the encouraging comments and for highlighting several important issues.
>
> **Quadratic assumption and its realism**
>
> We fully agree that the loss landscape is not globally quadratic in practice. Our use of quadratic losses is explicitly local: as stated in Section 3.1 and Appendix C, any twice-differentiable loss can be approximated as quadratic in a small neighborhood of a point $\theta$.
>
> The role of Theorems 1 and 2 is to describe what happens in such local neighborhoods when the train/test losses are modeled as random quadratics. GAM then applies these insights locally and adaptively along the optimization path, repeatedly re-estimating gradients and perturbations as $\theta$ moves.
>
> **Why require $f'(0) = 1$ in Theorem 2**
>
> The condition $f(0)=0$, $f'(0)=1$ reflects the intuition that very flat directions remain flat between train and test. If an eigenvalue is essentially $0$ in the training loss (extremely flat), we do not expect the test loss to suddenly become highly curved along that direction. In other words, parameter values achieving near optimal loss on the training set are also expected to achieve near optimal loss on the test set. Please see Assumption 9 in Appendix C for further discussion.
>
> **Approximation quality in Figure 4**
>
> You are right that in the toy example the learned transform does not perfectly match the true one. There are two main reasons:
>
> - We restrict ourselves to a finite-order polynomial $P(\lambda) = \lambda \sum_{t=1}^T \gamma_t \lambda^{t+1}$. By the Weierstrass theorem, we can approximate any continuous $f$ arbitrarily well on a bounded interval as $T\to\infty$, but in practice we keep $T$ small to control compute and numerical stability.
>
> - The transform only needs to be accurate on the range of eigenvalues actually present in $\tilde M$; behavior far outside this range is irrelevant for the optimization problem at hand.
>
> As we increase $T$, the approximation improves but cost grows. The hyperparameter study in Fig. 6 shows exactly this trade-off: large $T$ can become unstable/expensive, small $T$ is more robust but less expressive.
>
> **Gradient with respect to $\gamma_t$ (Algorithm 1, line 13)**
>
> Implementation-wise, $g_\gamma = \frac{\partial}{\partial (\gamma_1,\dots,\gamma_T)} \Delta(g_\theta,\bar g_\theta)$ is computed via standard automatic differentiation. The key is that $\hat\theta$ depends linearly on $\gamma_t$:
>
> $\hat\theta = \theta + \sum_{t=1}^T \gamma_t d^t(\theta)$,
>
>  so gradients can be back-propagated through $\hat\theta$ efficiently.
>
> **Experimental scale, accuracy levels, and additional results**
>
> We agree that ResNet-50/ImageNet is considered a relatively simple setting now and our accuracies are not state-of-the-art. We chose these benchmarks because they are still standard in the optimizer literature and make it feasible to run multiple methods and ablations with limited compute.
>
> Nevertheless, our goal is not to claim state-of-the-art *absolute* performance, but to show that (i) GAM empirically realizes the test-loss gradient transformation predicted by our theory, and (ii) this proposes an alternative explanation for the mechanism behind the generalization of SAM algorithms.
>
> To address scale concerns, we are currently training a GPT-2–style model on The Pile with GAM vs. SAM/SGD. If accepted, we will include these larger-scale results in the camera-ready.
>
> We already include hyperparameter sensitivity for GAM in Fig. 6 (effects of $T$, batch size, and $\epsilon$).
>
> Regarding Figure 2, SAM's curves are meant to show that regardless of the value of $\gamma_1$, SAM's transformation curve cannot match the sharper curve exhibited by GAM; the exact value of $\gamma_1$ is irrelevant for this message.
>
> Once again, we appreciate your careful reading and the helpful suggestions on both theory and experiments.

---

> > ### Comment · Reviewer_NWzY · 2025-11-20
> >
> > Thanks for the detailed rebuttal. However, due to the unrealistic quadratic assumption (authors do not provide enough evidence) and the under-scored accuracy on ImageNet-1K (far below 70% for SGD), I am not fully convinced that GAM would perform as well as claimed. So, I would like to maintain the score.

---

> > > ### Author Response · Authors · 2025-11-20
> > >
> > > Thank you for your response.
> > >
> > > Regarding the quadratic assumption, we again emphasize that any smooth function can be approximated as locally quadratic within a certain radius: this is a mathematical fact. We kindly ask the reviewer what kind of evidence they are looking for.
> > >
> > > Regarding the ImageNet results, we again emphasize that our goal is not to show state-of-the-art results, but rather to shine light on the mechanisms behind SAM algorithms. We do not claim that GAM is a state-of-the-art method.

---

> > > > ### Comment · Reviewer_NWzY · 2025-11-21
> > > >
> > > > I appreciate the authors for the quick reply. Unfortunately, they do not address my concerns:
> > > >
> > > > - **local quadratic assumption**. This is the cornerstone of their story. I admit that that smooth function can be approximated as locally quadratic within a certain radius. The first question is how to make sure that the loss function of a neural network is smooth. Second, even if the loss function is smooth, how large should be the radius to ensure the approximation holds. And empirically, if the approximation is reasonable, the authors should at least deomonstrate **the Hessian eigenvalues are non-negative** during different stages of training (e.g. select serval epochs from begining、midddle、end of training).
> > > >
> > > > - **low top-1 accuracy on ImageNet-1k**. I understand that GAM is not a state-of-the-art method and this is acceptable to me. However, the issue is that the reported baselines is too low. For example, for ResNet-50 trained by SGD, the authors only report an accuracy of 64.92%, while the popupar baseline at least reaches 76% (e.g. see https://docs.pytorch.org/vision/main/models.html).
> > > >
> > > > Given these two critical issues, I am afraid that I cannot raise the score.

---

> ### Author Response · Authors · 2025-11-21
> **Thank you for your continued engagement**
>
> Regarding whether loss functions of neural networks are smooth, note that the composition of any smooth functions is smooth. Thus, as long a network has a smooth nonlinearity (such as tanh), and the final loss function is smooth (such as softmax cross-entropy), then the loss landscape as a function of parameters is smooth.
>
> Regarding the size of the radius, as we understand it, the reviewer is concerned that in practice, the size of a radius for which the quadratic approximation holds is unreasonably small. To this point, we note that many proposed optimizers for deep networks assume a locally quadratic loss landscape to motivate their methods: examples include AdaHessian, Shampoo and K-FAC. In fact, popular and effective adaptive first order methods like Adam assume a locally *linear* loss landscape, even though this holds for an even smaller radius than a locally quadratic assumption. To be more constructive, we ask the reviewer, would it be sufficient if we conducted experiments empirically measuring the radius for which a quadratic approximation holds (within some fixed error $\epsilon$)?
>
> Regarding the Hessian eigenvalues, note that Theorem 1 does not enforce the Hessian eigenvalues to be non-negative. We apologize if any part of our paper suggested the Hessian must be positive semi-definite: we are happy to correct this. In fact, we don't expect the Hessian to be positive semi-definite except towards the end of training.
>
> Regarding the low accuracy on ImageNet, we believe the lower accuracy is due to us training the ResNet-50 for only 20 epochs. We limit training to thus number of epochs due to computational cost. However, if the reviewer is concerned that the improvements from GAM will degrade with more epochs, please note that in Figure 5, we show that for other networks, the accuracy gains from GAM are consistent, if not growing, through training.

---

> > ### Comment · Reviewer_NWzY · 2025-11-21
> >
> > First, due to the exisistence of ReLU and BN layers, the loss function of deep neural networks is not always smooth as expected. So, the validity of quadratic approximation should be carefully examined. At least, the authors should provide some empirical results.
> >
> > Second, regarding the Hessian eigenvalues, I made a mistake and am sorry for this.
> >
> > At last, it should be noted that an optimizer may perform dramatically different on small-scale and large-scale datasets. Therefore, the authors should provide relevant results on ImageNet-1K.

---

> > > ### Author Response · Authors · 2025-11-21
> > > **Thank you for your continued engagement**
> > >
> > > Regarding the point on architecture, we do not use ReLU in our experiments: we use the smooth softplus activation instead. We believe batch normalization is a smooth operation as well. The reviewer asks for empirical results, so we ask again: would it be sufficient if we conducted experiments empirically measuring the radius for which a quadratic approximation holds (within some fixed error $\epsilon$)?
> > >
> > > We completely agree that an optimizer may perform differently on large-scale datasets. In fact, to address these concerns, we are currently training a GPT-2–style model on The Pile with GAM vs. SAM/SGD and will present these results once available.

---

### Official Review · Reviewer_3dqg · 2025-10-25

**Soundness:** 3
**Presentation:** 3
**Contribution:** 3
**Rating:** 6
**Confidence:** 4

**Summary:**

This paper addresses the unclear mechanism by which the SAM optimizer improves generalization. It proposes a novel optimizer, GAM, which directly optimizes the expected test loss. The authors prove that the true expected test loss is a rescaled version of the training loss. Building on this, GAM employs a sequence of perturbed updates to directly optimize the expected test loss. Furthermore, the authors design a practical online algorithm that outperforms SAM on several benchmarks. This work demonstrates strong theoretical rigor, innovative methodology, and robust empirical support, with significant heuristic value.

**Strengths:**

- Provides a solid theoretical foundation for understanding the generalization capability of the SAM optimizer.
- Reveals that a sequence of parameter perturbations can transform the gradient of the training loss into that of the test loss, bridging the gap between optimizing training loss and directly optimizing expected test loss.
- Well-structured and easy to read.

**Weaknesses:**

Although the proposed method is theoretically sound, it incurs substantial computational overhead, limiting its practical applicability.

**Questions:**

- In equation 11, should the second instance of  $D^t$ be $D^{t-1}$?
- It is unclear how line 8 in Algorithm 1 is derived from equation 11; further derivation details are needed.
- The experimental results in Table 1 lack performance data for CRSAM on the ImageNet dataset; an explanation is required.
- The process for generating Figure 2 is not sufficiently detailed; more methodological description is recommended.

---

> ### Author Response · Authors · 2025-11-19
>
> Thank you for the positive assessment and for the concrete questions.
>
> **Compute overhead**
>
> We agree that a $3\times$ overhead relative to SGD is substantial. Our primary goal in this work is **not** to propose the most efficient SAM-style optimizer, but to **clarify the mechanism** by which SAM-like perturbative methods can approximate the expected test loss. There is a rich literature on making SAM/second-order methods cheaper (e.g., approximations to Hessians and to SAM’s inner maximization); our contribution is complementary and more conceptual.
>
> **Equation (11)**
>
> Thank you for catching this typo! We have fixed the typo in the revised manuscript.
>
> **Algorithm 1, line 8**
>
> Line 8 in Algorithm 1 is just a finite difference approximation of the derivative in Equation (11). $\zeta$ is set to values $\epsilon$ and $0$ for this finite difference approximation. That is, we approximate:
> $\frac{\partial}{\partial \zeta} D^{1}(\theta + \zeta D^{t-1}(\theta)) |_{\zeta=0} \approx \frac{1}{\epsilon} [D^{1}(\theta + \epsilon D^{t-1}(\theta)) - D^{1}(\theta)]$
>
> **CRSAM on ImageNet**
>
> We did not include CRSAM on ImageNet due to compute constraints. Nevertheless, recognizing the need for more experimental validation, we are currently training a GPT-2–style model on The Pile with GAM vs. SAM/SGD. If accepted, we will include these larger-scale results in the camera-ready.
>
> **How Figure 2 is generated**
>
> For each training method, we estimate the effective eigenvalue transform $f$ induced by its update rule. Note that following Theorem 2, $f(x)$ is approximated as $f(x) = x + \sum_t \gamma_t x^{t+1}$. Thus, for any given training method, we convert its sequence of $\gamma$ values into a function $f$ and plot it. For example, for SGD, $\gamma_1=0$, so the corresponding $f$ is the identity function.

---

### Official Review · Reviewer_Zqnc · 2025-10-27

**Soundness:** 2
**Presentation:** 3
**Contribution:** 2
**Rating:** 2
**Confidence:** 4

**Summary:**

This paper aims at better generalization of neural network by directly optimizing on the testing loss landscape. This is done by first theoretically showing the testing loss landscape is a rescaled version of the training loss landscape. Then the authors show such difference can be covered by perturbative steps, arriving at GAM algorithms that using multiple online-adjusted perturbative steps. Experiments show GAM surpasses SGD and SAM. It also implies SAM, as single-step GAM, can be seen as implicitly optimizing directly on the testing loss landscape.

**Strengths:**

- The paper features theoretical analysis that reveals the connection and difference between the testing and training loss landscape.
- Such connection is used to develop algorithm that optimizes directly on testing loss landscape, which is well motivated.
- Experiments demonstrate that GAM surpasses SGD and SAM with statistical significance.
- SAM is connected to GAM, revealing its implicit role of simulating testing loss landscape. This connection provides a new perspective on understanding SAM.

**Weaknesses:**

- Gap between theoretical results and aim / practical use: Theorem 1 lay down the theoretical foundation for this paper. However, its conditions are a bit difficult to comprehend at first glance, especially with quadratic approximations involved. For example, the randomness of quadratic loss parameters ($\\tilde{\\theta}, \tilde{M}, \dots$) is abstracted / comes from which part of the whole training? What does the whole training process as a random process, where the loss parameters are involved, look like. Therefore, I suspect that such result cannot be directly applied to the training process. For example (please correct me if I embed Thm 1 into the training process in a incorrect way),  the training process is modeled by the following random process:

  $$
    (\bar{X}, \bar{Y}) \overset{\text{training}}{\rightarrow} \theta \underset{\text{training loss}}{\overset{\text{approximating}}{\rightarrow}} (\tilde{\theta}^*, \tilde{M}, \tilde{c})
  $$

  and at the same time

  $$
    \theta \underset{\text{testing loss}}{\overset{\text{approximating}}{\rightarrow}} (\theta^*, M, c).
  $$

  Noting that $(\\tilde{\\theta}^\*, \\tilde{M})$ and $(\\theta^\*, M)$ have a confounder $\theta$, they are correlated and one cannot have $\\theta^\*, \\tilde{\theta}^\* \\perp  M, \\tilde{M}, c, \\tilde{c}$, which is an assumption of Theorem 1. Intuitively, it says the training may bias the parameter to empirical-loss flat minima, where the testing-loss flatness may differ a lot, just like overfitting. I believe it is indeed the case especially when the parameter is explicitly or implicitly (eg, SGD is well-known to have flatness bias) optimized toward flat minima.
- Extra sample use: In Line 5, Algorithm 1, new data is directly sampled from the data distribution to measure how much $\\bar{g}\_{\theta}$ deviates from $g_{\theta}$. I read Appendix E and found no indication about how this data is sampled (so I assumed it is some new data. Correct me if I was wrong) and how much of them is used. If this data is indeed newly sampled, I suspect severely unfair comparison in experiments where SAMs do not use extra data. If it is reused training data, then this data is already used in training and is (over)fitted. Can it estimate the testing data without bias? If it comes from training data partition, then will it hurt data efficiency?
- Such direct access to testing loss landscape also leaves Theorem 1 not fully exploited. Abstractly, Theorem 1 involves by revealing that the difference between the empirical and testing loss landscape has a rescaling-like structure and Theorem 2 says some perturbative can cover this gap, but how it is covered is by directly comparing the two landscapes instead of exploiting the explicit difference revealed by theoretical results.
- Experiments: The experiments use networks whose architectures may be too old. Also, for deeper networks, GAM seems unable to surpass CRSAM. ImageNet results for CRSAM is also missing.

**Questions:**

1. In the paper, Theorem 2 helps constructing the true testing loss gradients by moving the parameter toward a new place where the empirical gradients have similar gradients with testing gradients, i.e., simulating the gradient modification by moving where the gradient is computed. What is the motivation behind this simulation? Why not just do some calculation and moving the empirical gradients instead? I found "surprising similarity between the update mechanisms of SAM and GAM" in the Discussion. Is it related to unifying SAM with GAM?

---

> ### Author Response · Authors · 2025-11-19
>
> Thank you for the detailed comments and for engaging carefully with both the theory and the algorithm.
>
> **How to interpret Theorem 1 and its assumptions**
>
> As the reviewer notes, due to the quadratic approximation potentially changing at different locations in the loss landscape, $L$ and $\tilde L$ have a confounder $\theta$. This means that indeed, the independence condition $\theta^\star, \tilde \theta^\star \perp M, \tilde M, c, \tilde c$ may not hold exactly.
>
> Intuitively, this assumption states that the curvature axes (eigenvectors) of the quadratic and the location of the minimizer do not have a privileged alignment. In other words, the direction of the optimum is not “special” relative to the curvature directions. This we believe is a relatively reasonable symmetry assumption.
>
> Contrary to what the reviewer claims, this does **not** assume that train- and test-flatness match; the whole point of Theorem 1 is that the eigenvalues **do change** (via $D(\tilde\Lambda)$).
>
> **Use of the “extra sample” in Algorithm 1 (line 5)**
>
> We clarify that in the actual implementation, the “auxiliary” minibatch $(\bar X, \bar Y)$ is not fresh held-out data; it is simply the previous training minibatch stored in memory. That is, GAM does not use extra data beyond what SAM/SGD see.
>
> Concretely, when a new minibatch $(X,Y)$ arrives, we use the most recent past minibatch as $(\bar X,\bar Y)$ and then overwrite it at the end of the iteration. This avoids data-efficiency issues while still providing a stochastic proxy for the test gradient with a different sample than $(X,Y)$.
>
> However, as the reviewer notes, because we use the previous minibatch, our approximate test gradient is not an unbiased sample from the true data distribution. If we do wish to generate a bias-free, uniform sample from the data set, this would hurt data efficiency.
>
> **“Not fully exploiting” Theorem 1**
>
> We agree that GAM does not plug in a **closed-form** rescaling from Theorem 1. Theorem 1 tells us that
>
> - the expected test Hessian shares eigenvectors with the train Hessian, and
>
> - there exists some diagonal mapping $D(\tilde\Lambda)$ of eigenvalues, but computing $D$ exactly would require the conditional distribution of $M$ given $\tilde M$, which is intractable without strong extra modeling assumptions.
>
> Rather than making additional untestable assumptions, we use Theorem 2 and the perturbative construction to learn an effective spectral transform from data via $\gamma_t$. In this sense, Theorem 1 is used structurally (to justify an eigenvalue-only transformation), while Theorem 2 plus GAM provide a practical gradient-oracle-based way to approximate that transformation.
>
> **Concerns on experimental setting**
>
> We agree that ResNet-50/ImageNet is no longer considered state-of-the-art. We chose these benchmarks because they are still standard in the optimizer literature and make it feasible to run multiple methods and ablations with limited compute. Due to the computational cost of running CRSAM on Imagenet, we omit it from our experiments.
>
> Our goal is not to claim state-of-the-art *absolute* performance, but to show that (i) GAM empirically realizes the test-loss gradient transformation predicted by our theory, and (ii) this proposes an alternative explanation for the mechanism behind the generalization of SAM algorithms.
>
> To address scale concerns, we are currently training a GPT-2–style model on The Pile with GAM vs. SAM/SGD. If accepted, we will include these larger-scale results in the camera-ready.
>
> **Motivation behind Theorem 2**
>
> The motivation behind Theorem 2 is exactly to show that SAM-like perturbations can implement a desired spectral transform of the Hessian using only a gradient oracle, without explicitly computing or storing the Hessian.
>
> Directly “moving” the gradient by applying a matrix transform $f(\tilde\Lambda)$ would require eigendecomposing the Hessian or at least approximating it at large scale, which is prohibitive for modern networks. In contrast, Theorem 2 shows that one can approximate the same effect by composing directional derivatives $D^t(\theta)$ along gradients, i.e., via a sequence of perturbation-and-regradient steps- precisely the mechanism SAM already uses.
>
> This is what we meant by the “surprising similarity” between SAM and GAM: SAM corresponds to $T=1$ and a fixed $\gamma_1$, while GAM uses multiple learned $\gamma_t$ to implement a richer eigenvalue transform. This gives a new lens on SAM’s success: it can be seen as a low-order approximation to the test-loss gradient transformation embodied more faithfully by GAM.

---

> ### Comment · Reviewer_Zqnc · 2025-11-20
>
> Thank you very much for your detailed clarification!
> The rebuttal has provided useful clarification on the source of "extra" data, the experimental setting and the SAM-similarity aspects.
>
> However, I still have some questions regarding the rest, which, to some extent, can be summarized as whether the motivation and the claimed goal of getting across the gap between the training and testing loss landscape are indeed achieved.
>
> Regarding Theorem 1, the author has provided the intuition behind the independence condition. Is it possible to provide a rigorous version of the assumption? Regarding the training-testing curvature match/change, although Theorem 1 indeed states the testing and training curvature mismatches in the sense of eigenvalues, it stills states a **directional** match. I still suspect this directional match, given the explicit/implicit second order optimization. It seems easy to construct a case where curvatures of every (or the only) local minima at the training and testing landscapes directionally mismatch. For example, consider a learning task whose sample space has only two samples and training set is of size $1$. Each of them correspond to a vector $v_i$, which induces the loss to be $L_i(x) = x^\top v_i v_i^\top x$. As long as $v_1, v_2$ have equal norms and are neither orthogonal or parallel, then testing loss's curvature $v_1 v_1^\top + v_2 v_2^\top$ has eigenvectors (with different eigenvalues so they are unique) $\\{\frac{v_1 + v_2}{2}, \frac{v_1-v_2}{2}\\}$, which do not directionally match with curvatures of $L_1, L_2$. Does this case serves as a counter-example, or it is ruled out by some other assumptions of interest?
>
> Regarding the learning of $\gamma_t$, it still requires testing data or proxies. If proxies are used, then overfitting infiltrates int o the learning of $\gamma_t$. So the gap is still not got across because fresh new data is still required.
>
> Lastly, is it possible to directly check whether the goal of optimizing directly on the testing landscape is achieved? For example, displaying the training loss with the testing loss. Or, comparing the estimated testing loss gradients (or the gradients of local second-order approximation of testing loss) with the gradients found the GAM?

---

> > ### Author Response · Authors · 2025-11-20
> >
> > Regarding the Theorem 1, the reviewer is correct that the testing and training curvatures have a directional match. We clarify that, this is not an assumption, but rather the *result* of Theorem 1. Regarding the supposed counterexample proposed by the reviewer, the key is that Theorem 1 is a statement about the *expected* test loss (conditioned on observing the training loss) rather than the actual test loss. If we indeed only observe loss $L_1$ during training (which has curvature $||v_1||^2$ in the $v_1$ direction and curvature $0$ in all orthogonal directions), then by symmetry we would expect the test loss to directionally have the same curvature structure: specifically, some fixed curvature in the $v_1$ direction and some other curvature in all orthogonal directions. Critically, we *cannot* expect to know $v_2$ since $L_2$ not observed during training. In fact, if we assume $v_2$ has a spherically symmetric distribution, then the expected test loss (having observed $L_1$) is $x^T (v_1 v_1^T + \lambda I) x$ for some constant $\lambda$ which has eigenvector $v_1$  (with eigenvalue $||v_1||^2 + \lambda$) and eigenvectors in all orthogonal directions (with eigenvalue $\lambda$).
> >
> > Regarding the learning of $\gamma$, yes, we agree that because we use a proxy rather than fresh new data, our approach is susceptible to overfitting. We admit this as a limitation of our method, although empirically, the overfitting is not severe enough to make GAM perform worse than SAM.
> >
> > Regarding validating whether we indeed optimize directly on the test landscape, this is in fact precisely the goal of Figure 4 which shows that GAM can in fact approximately recover the correct transformation from train to testing landscape. As for comparing the testing loss gradients with the gradients found by GAM, note that the dot product between the test loss gradient and the GAM gradient is precisely the change in test loss at each time step. Figure 5 shows the test set performance trajectories over the course of training, revealing that the GAM gradient is most aligned with the test loss early in training.

---

> > > ### Comment · Reviewer_Zqnc · 2025-11-21
> > >
> > > Thank you very much for your further clarification.
> > > However, I must clarify that the proposed counter-example is directly targeted at the **result** *instead of assumption* of Theorem 1. It shows a case where training and testing curvature cannot directionally match at all, which is directly against the result of Theorem 1 stating that they can directionally match. This shows that there seems something wrong when one tries to apply Theorem 1 to actual training. Although the difficulty may indeed comes from the assumption, the counter-example on the result may suggests one cannot find any assumption that implies directional match (or similar expected form) and can be applied to actual training.
> > > Regarding the expectation, the construction uses a constant testing landscape, whose expectation conditioned on anything is itself.
> > >
> > > Regarding the experiment displayed in Figure 4, I apologize for not paying enough attention to it. However, the result is based on synthetic data. I believe direct validation on real neural networks and natural data would be more convincing. Specifically, Appendix D only states how to construct the observed training curvature eigenvalues, so I must assume a rescaling-like relation between the testing and training curvature (please correct me if incorrect). Therefore, the experiment in Figure 4 is more like verifying GAM and Theorem 2 assuming the result of Theorem 1. I fully agree that Theorem 2 and GAM can achieve the goal of recovering testing curvature **given** the curvature rescaling relation stated by Theorem 1 and it is a nice discovery that using high-order derivatives can cover the rescaling relationship, but they rely on the result of Theorem 1 and I am not convinced on whether the relation stated by Theorem 1 indeed holds true in training under deep NN and actual data. Let alone a lot of critical differences between the synthetic setting and the actual training that potentially makes the directional match easier, eg, constant curvature in each loss landscape and only one valley, which forbid any form of "overfitting" on flatness. Therefore, it is more convincing to display results on the practical instead of synthetic settings.

---

> ### Author Response · Authors · 2025-11-21
> **Thank you for your continued engagement**
>
> Regarding your proposed counterexample, we fully agree that the training and testing curvature directions do not match. Again, we emphasize that Theorem 1 is not about the relationship between the *true* testing and training curvature directions, but rather the *expected* testing and training curvature directions. Given only $v_1$, by symmetry, the expected test loss landscape *must* have the structure $x^T(\alpha v_1 v_1^T + \beta I) x$ since we have no prior information about $v_2$. This is precisely what Theorem 1 shows. We kindly ask the reviewer to clarify how this constitutes a counterexample.
>
> Regarding the reviewer's second point, if we understand correctly, the concern is whether Theorem 1 would hold in practice for deep neural networks on real data. This unfortunately is difficult to directly evaluate empirically because of the reason above: in practice, we only have access to the *true* testing and training landscapes, not the *expected* testing landscape conditioned on observing the training data. Just as your counterexample showed, we would find that the curvature directions of the *true* testing and training landscapes do not match in practice. This is why we evaluate Theorem 1 indirectly by proposing GAM and measuring its performance. If the reviewer can suggest an alternative evaluation to directly measure the similarity between the *expected* testing and training landscapes on real data, we are happy to try it.

---

> ### Comment · Reviewer_Zqnc · 2025-11-22
>
> Thank you very much for your further feedback.
>
> Now I see I might have some misunderstanding on the "expected loss landscape".
> I am not quite familiar with Bayesian optimization. Is this expected loss landscape something like the Gaussian process posterior of Bayesian optimization that models unknown with some prior (your symmetric $I$ curvature?), or the expectation over the prior over the true loss landscape? I previously assumed the latter and in this case, I could set the prior to be the Dirac delta distribution on $L(x) = x^\top (v_1 v_1^\top + v_2 v_2^\top) x$. As a result, the testing loss landscape is a constant random variable and the expected loss landscape is itself, with mismatch eigenvectors. Although this example has trivial prior, one can add small random perturbation to $v_1, v_2$ to construct a non-degenerate prior.
> After you explained the by-symmetry $\dots+\beta I$ expected loss landscape, it seems it is the former or something in between that replaces unknown parts of the true loss landscape using some prior distribution.
>
> So could you please further clarify which notion is the one used? If it is the former, then how does expected loss landscape relate to the true loss landscape and help optimizing the latter? It seems the two can be drastically different, like the $+ \beta I$   Also, it seems the pseudo code of GAM also learns $\gamma_t$ by estimating the true loss landscape. Is my understanding correct and how does it relate to optimize on expected testing loss landscape?
>
> Also, I am quite confused on how exactly Theorem 1 applies to training process to provide foundation for Theorem 2 and GAM that targeted to NN training. Could the author provide the official interpretation of Theorem 1 on this, like how to define the training process as a random process so that variables in Theorem 1 show up? This may also helps clarify the expected loss landscape.

---

> > ### Author Response · Authors · 2025-11-22
> > **Official Comment by Reviewer Zqnc**
> >
> > Regarding the interpretation of "expected loss landscape", we clarify that the expected testing loss landscape is the *posterior* of the testing loss landscape after having observed the training loss landscape. As the reviewer correctly notes, if the prior distribution on the testing loss was set to a Dirac delta distribution on $L(x) = x^\top (v_1 v_1^\top + v_2 v_2^\top) x$, then the posterior is also the same Dirac delta distribution. This indeed leads to a mismatch between the testing and training loss eigenvector directions. However, our symmetry assumptions in Theorem 1 explictly rule out such a prior distribution: namely, equation 5 enforces that there cannot be "preferred" directions in the testing loss landscape. Thus, the reviewer is correct that we "replaces unknown parts of the true loss landscape using some prior distribution".
> >
> > Indeed, the expected test loss landscape can be dramatically different than the true test loss landscape. As the reviewer correctly points out, GAM uses the *true* test loss landscape to learn the transformation between the training and *expected* test loss landscape. This is because the expected test loss landscape is not directly accessible, thus we must use the true test loss landscape as a proxy to learn the appropriate transformation.
> >
> > Regarding the relationship between Theorem 1 and the training process, Theorem 1 states the relationship between the training loss and the expected test loss: namely that they have the same Hessian eigenvector directions. During training, we have access to the training loss landscape. SGD naively optimizes directly on the training loss landscape. GAM uses a different approach: rather than directly optimizing on the training loss landscape, it aims to optimize on the expected test loss landscape. However, the expected test loss landscape is not directly accessible. Fortunately, Theorem 1 indicates that the expeted test loss landscape is simply a "rescaled" version of the training loss landscape. Therefore, GAM estimates these "rescaling" factors online and then optimizes directly on the estimated expected testing loss landscape.

---

### Official Review · Reviewer_DkRZ · 2025-11-01

**Soundness:** 3
**Presentation:** 4
**Contribution:** 3
**Rating:** 4
**Confidence:** 3

**Summary:**

The paper derives a relationship between the expected test loss and the training loss: they share eigenvectors but the eigenvalues differ, so that the Hessian is a rescaling. From this, they come up with their Generalization-Aware Minimization algorithm that estimates this. On CIFAR-10/100, SVHN and ImageNet, they find that at the same iteration, GAM outperforms SAM and

**Strengths:**

I'm plausibly unaware of work in the past couple years that could've made this seem more obvious, but the rescaling result seems interesting on its own.

I appreciated adding some intuition and explanation of lines in the algorithm, and the way the assumptions are laid out and justified in §C; note I have not checked the proofs in much detail.

The figures were also good: Figure 1 was very clear and Figure 2 was valuable for further connecting the theory and the outcomes.

**Weaknesses:**

I'm not sure I would call 3x "manageable" overhead; at the very least, I expect to see compute-matched results somewhere, for those applications where that's more of a constraint than data (like pretraining foundation models). Saying "Future work
could explore approximations or scalable implementations of higher-order derivatives" felt weak to me, given all the work out there on approximating second-order derivatives already.

The settings feel a little unrealistic in 2025, maxing out at a ResNet-50 on ImageNet.

The gains also seem small for that overhead, e.g., 0.7% on ImageNet (when the results are in the 65% range, not near saturation)

**Questions:**

I was confused by Equation 11; where's the recursion? Shouldn't there be a t-1 like in line 8 of Algorithm 1?

---

> ### Author Response · Authors · 2025-11-18
>
> Thank you for the thoughtful and constructive review.
>
> **Compute overhead and “manageable” cost**
>
> We agree that a $3\times$ overhead relative to SGD is substantial. Our primary goal in this work is **not** to propose the most efficient SAM-style optimizer, but to **clarify the mechanism** by which SAM-like perturbative methods can approximate the expected test loss. There is a rich literature on making SAM/second-order methods cheaper (e.g., approximations to Hessians and to SAM’s inner maximization); our contribution is complementary and more conceptual.
>
> **Scale and realism of experimental setting**
>
> We agree that ResNet-50/ImageNet is no longer “state-of-the-art scale” in 2025. We chose these benchmarks because they are still standard in the optimizer literature and make it feasible to run multiple methods and ablations with limited compute.
>
> Our goal is not to claim state-of-the-art *absolute* performance, but to show that (i) GAM empirically realizes the test-loss gradient transformation predicted by our theory, and (ii) this proposes an alternative explanation for the mechanism behind the generalization of SAM algorithms.
>
> To address scale concerns, we are currently training a GPT-2–style model on The Pile with GAM vs. SAM/SGD. If accepted, we will include these larger-scale results in the camera-ready.
>
> **Equation (11) recursion**
>
> Thank you for catching this typo! We have fixed the typo in the revised manuscript.

---

> ### Comment · Reviewer_DkRZ · 2025-11-20
> **Help me understand**
>
> Thanks for the reply. I'm wondering if it could be helpful to spell out (here, not in the paper necessarily) what you speculate could follow from the conceptual contributions of the paper, if you don't want it to be measured as much as a practical optimizer alternative; the field is littered with attempts to beat Adam, for example, so assuming GAMv3 or something would find a niche like Shampoo (for example, and that's arguable, even) is not a given.
>
> I agree that ResNets are common here, but is there nothing you can share about your GPT-2 results before December 3? I'm not a fan of holding results hostage for camera-ready, when of course there's little hope of us seeing or saying anything at that point (I say, even though we've done the same).
>
> Currently I plan to follow the other discussions and see where they go; of course, a lot of my comments were also made there, and then some.

---

> > ### Author Response · Authors · 2025-11-20
> > **Thank you for your thoughtful comment**
> >
> > We'd like to place the paper as follows: rather than proposing a new, superior optimizer, we would like our main contribution to be proposing an alternative explanation for why SAM-like algorithms generalize well. In particular, we hypothesize that the gradient transformation perspective could be a more useful and accurate explanation of why SAM-like algorithms generalize well rather than the usual sharpness/flatness explanation. In contrast to other papers which explain why SAM generalizes well (e.g. Wen et al., 2023; Andriushchenko & Flammarion, 2022), we validate our hypothesis on why SAM might work by propsing an alternative optimizer, GAM, that empirically performs better. Of course, we don't claim to have fully solved the question of why SAM algorithms generalize well, merely propose a plausible alternative explaination with some preliminary empirical evidence. Nevertheless, we hope this can shine light on the theoretical underpinnings of SAM.
> >
> > Regarding the GPT-2 results, we will try our best to post some preliminary results before Dec 3rd, but given the computational cost of training this model, the results would be more meaningful given more training time.

---

### Meta-Review · Area_Chair_muxG · 2025-12-26

**Summary:**

introduces an enhanced approach to the Sharpness-Aware Minimization (SAM) optimizer, which is utilized to improve neural network generalization. The authors present a theoretical connection between the training loss landscape and the expected test loss landscape, showing that the expected test loss landscape is a rescaled version of the training loss landscape. They propose an algorithm that employs perturbative updates to optimize the expected test loss directly. Empirical results demonstrate that GAM outperforms SAM on several benchmarks, providing insights into how sharpness-based optimizers enhance generalization.

I have read the paper and all the discussions. The reviewers and I share the following concerns.

Computational Overhead: Reviewers and I noted that the overhead of GAM relative to SGD was substantial, raising concerns about its practical applicability and feasibility for large models.

Experimental Validation: The experiments primarily utilized ResNet-50 on standard datasets. Reviewers expressed concerns about the realism of these settings, suggesting that they may not adequately represent modern benchmarks or scale.

Quadratic Assumption: There were significant concerns regarding the assumption that the loss landscape is locally quadratic. Reviewers questioned how this assumption holds in practice, given the presence of various nonlinearities in neural network architectures.

Representation of Testing Loss: Some reviewers expressed skepticism about the claimed ability of GAM to directly optimize the expected testing loss landscape, suggesting a potential mismatch between the training and testing loss landscapes.

**Reviewer Concerns:**

The authors acknowledged the substantial computational overhead and clarified that their primary contribution is to provide a conceptual framework, not necessarily to design the most efficient optimizer. I do not think that this concern was adequately addressed because the paper seems to position itself as proposing another optimizer that possibly improves generalization. However, I do not see any results that claim to improve generalization.

The response to concerns about experimental settings indicated that they were in the process of training larger models, such as GPT-2, to provide further validation of GAM. This would be good to include in a future version of the paper; however, accepting the paper in the current form would be considered premature.

In regard to the quadratic assumption, the authors explained the local smoothness of the loss landscape in neural networks and stated they would conduct empirical evaluations to assess the validity of the quadratic approximation. Personally, as the AC, I feel that this assumption oversimplifies the analyses. Most loss landscapes are not quadratic, but can be approximated by quadratics locally. Yes, this is true, but what is critical is the behavior of the optimizer to work well globally. This is scarcely discussed in this paper.

They also clarified the relationship between the training and testing loss landscapes and emphasized that GAM utilizes a transformation that is influenced by the underlying theory. However, I believe this is still a hypothesis on the part of the authors.

**Reviewer Scores:**

Overall, after reading the discussions, which were rather extensive, I believe that the reviewers still had several lingering issues with the paper. I do not believe that they would be inclined to raise their scores. Hence, despite the authors’ efforts to clarify and respond to the reviewers' concerns, the following reasons support a rejection decision:

High Computational Overhead: The substantial overhead associated with GAM, as noted by reviewers, raises concerns about its utility in practical applications, especially in light of the ongoing development of more efficient optimizers within the community.

Limited and Outdated Experimental Validation: While the authors indicated plans for larger-scale experiments, the current results remain limited to older architectures. Reviewers highly emphasized the need for more realistic and representative benchmarks to establish the effectiveness of GAM.

Questionable Theoretical Assumptions: The reliance on the quadratic assumption and doubts about its applicability in practice raise significant concerns regarding the robustness of the theoretical framework. Many reviewers suggested that the evidence provided did not sufficiently support the use of this assumption.

Unclear Impact of Expected Test Loss Optimization: The reviewers were not fully convinced that GAM effectively addresses the gap between training and expected test loss landscapes, questioning the practical application of the results derived from the theoretical claims.

---

### Decision · Program_Chairs · 2026-01-26

Reject